# LEARN OUT OF THE BOX:
# OPTIMIZING BOTH DIVERSITY AND PERFORMANCE IN OFFLINE REINFORCEMENT LEARNING

## ABSTRACT

In offline reinforcement learning, most existing methods have focused primarily on optimizing performance, often neglecting the promotion of diverse behaviors. While some approaches generate diverse behaviors from well-constructed, heterogeneous datasets, their effectiveness is significantly reduced when applied to less diverse data. To address this, we introduce a novel intrinsic reward mechanism that encourages behavioral diversity, irrespective of the dataset's heterogeneity. By maximizing the mutual information between actions and policies under each state, our approach enables agents to learn a variety of behaviors, including those not explicitly represented in the data. Although performing out-of-distribution actions can lead to risky outcomes, we mitigate this risk by incorporating the ensemble-diversified actor-critic (EDAC) method to estimate Q-value uncertainty, preventing agents from adopting suboptimal behaviors. Through experiments using the D4RL benchmarks on MuJoCo tasks, we demonstrate that our method achieves behavioral diversity while maintaining performance across environments constructed from both heterogeneous and homogeneous datasets.

## 1 INTRODUCTION

Learning to perform tasks with a range of behaviors, often referred to as quality-diversity optimization, is a growing area in reinforcement learning (RL) (Fontaine & Nikolaidis, 2021; Cully & Demiris, 2017; Pugh et al., 2016). Current methods for promoting diverse behaviors have primarily focused on online RL (Nilsson & Cully, 2021; Pierrot et al., 2022). These approaches, by encouraging policies to behave diversely, enhance environmental exploration (Hong et al., 2018) and facilitate skill discovery (Eysenbach et al., 2019; Sharma et al., 2020; Chen et al., 2024), ultimately leading to improved performance. However, producing diverse behaviors in offline RL (Kostrikov et al., 2021; Wu et al., 2020; Kumar et al., 2019; Fujimoto & Gu, 2021; Wang et al., 2020b) remains relatively unexplored and poses challenges due to the static nature of training datasets.

Learning diverse behaviors from offline data is just as important as in online scenarios. As the saying goes, "all roads lead to Rome," meaning multiple strategies can lead to successful outcomes, with each unique behavior in the dataset reflecting a distinct style or preference. Recent advancements in offline RL, such as SORL (Mao et al., 2024) and DIVEOff (Osa & Harada, 2024), have focused on learning diverse behaviors from heterogeneous datasets. These methods employ expectation-maximization (EM) algorithms to cluster data and train policies based on those clusters, fostering diversity in actions. While they effectively leverage dataset diversity to develop varied agent strategies, they often face challenges with non-heterogeneous datasets and tend to prioritize one aspect over another in the balance between quality and diversity optimization.

In this study, we introduce a unique behavior (UB) objective function designed to enable offline RL policies to learn diverse behaviors, *even from homogeneous datasets.* The objective is derived from maximizing mutual information between actions and policies within each state. This dependence between actions and policies naturally leads to distinct behaviors. Unlike baseline methods that achieve diversity through data segmentation, our method encourages agents – through a UB reward – to behave differently. This reward directly guides policy training, allowing agents to *learn actions not present in the training data.*

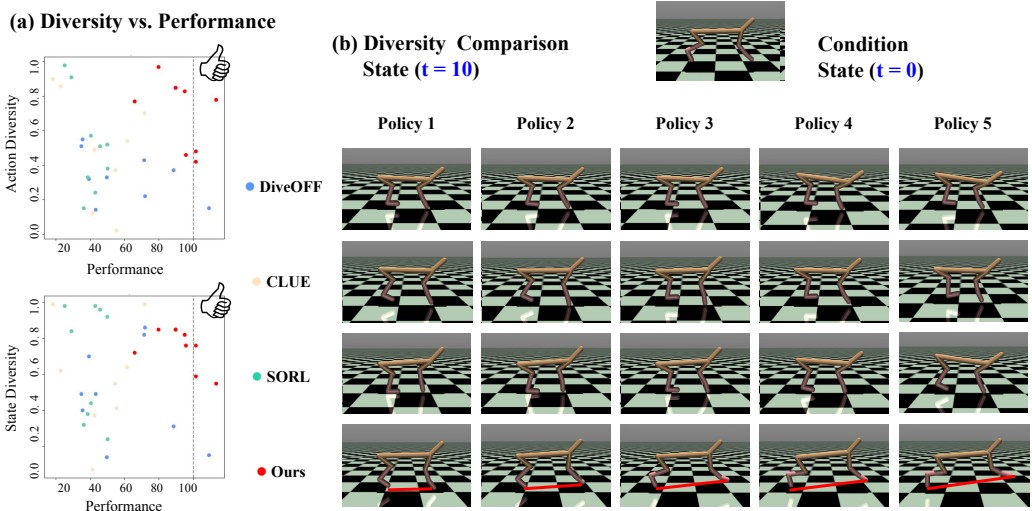

Figure 1: (a) The scatter plot illustrates the relationship between performance and diversity, in terms of action and state, across various methods and environments in the standard D4RL dataset. Each point represents a policy trained on a specific dataset, with different colors corresponding to different methods. Notably, **policies trained using our approach tend to cluster in the upper-right corner of the plot**, reflecting both high performance and diversity. (b) To visually compare the diversity between policies trained with baseline methods and our approach, we initialized the policies from a consistent initial state at $t = 0$ and rendered the resulting states after 10 steps. For additional visual comparisons, please refer to Figure 4 in the Appendix.

One major challenge in offline RL is handling out-of-distribution (OOD) state-action pairs, as rewards can only be estimated rather than obtained from the environment. Since our approach allows policies to deviate from the training data, it risks distorting learning outcomes. To address this, we adopt the Ensemble-Diversified Actor-Critic (EDAC) method (Kumar et al., 2020), which employs an ensemble of networks to estimate Q-values. By selecting the minimum Q-value from the ensemble, EDAC provides a more conservative Bellman target estimate, reducing the risk of overestimating Q-values for OOD state-action pairs. This ensures that while policies generate diverse actions, the quality of those actions is maintained.

We evaluated our approach against baseline methods using the D4RL (Fu et al., 2020) and diverse D4RL (Osa & Harada, 2024) datasets, which are rigorous benchmarks for offline RL algorithms. The results in Figure 1 and Table 1 show that, while there is typically a trade-off between performance and action/state diversity, our method achieves high performance while ensuring significant diversity across both homogeneous and heterogeneous datasets.

## 2    RELATED WORK

### 2.1    DIVERSITY IN OFFLINE RL

While multiple strategies can lead to successful outcomes, learning diverse behaviors from datasets contributed by multiple users can reflect their unique styles and preferences, which is beneficial in many real-world applications. The exploration of training diverse agents in offline settings has recently gained attention. For instance, CLUE (Liu et al., 2023) uses a Variational Autoencoder (VAE) to learn latent representations of state and action, defining different partial datasets as behavioral targets. It employs the latent distance from other data to the target behavior as an intrinsic reward, enabling the learning of varied behaviors. Similarly, SORL (Mao et al., 2024) uses an expectation-maximization (EM) algorithm, where the E-step estimates the posterior distribution of the latent variable based on current policies, and the M-step updates policies to maximize a lower bound of the posterior log-likelihood, achieving diverse latent behaviors. DIVEOFF (Osa & Harada, 2024) also applies the EM algorithm, augmented with a VAE for latent representation. In DIVEOFF's E-step,

the latent-conditioned policy is updated based on the posterior distribution of the latent variable, while the M-step updates the posterior distribution of the latent variable given the latent-conditioned policy. Unlike SORL, which uses policy probability density to compute the posterior, DIVEOFF uses a VAE to learn the latent posterior distribution given a state and action, enhancing learning by maximizing the mutual information between latent representations and state-action pairs.

Existing methods promote diversity by encouraging agents to learn from data that are distant in latent space. However, a challenge arises when the dataset is homogeneous, or when distant latent vectors are decoded into similar actions. To address this, we introduce an alternative approach with a diversity objective function defined in the action space. Agents are rewarded for behaving differently from each other during training, providing direct guidance that allows policies to learn behaviors even beyond the dataset distribution.

## 2.2 MUTUAL INFORMATION IN RL

Mutual Information (MI) has been widely used in online RL to enhance exploration through mechanisms such as empowerment (Klyubin et al., 2008; 2005; Mohamed & Jimenez Rezende, 2015; Leibfried et al., 2019) and information-theoretic curiosity (Still & Precup, 2012; Bai et al., 2021; Tao et al., 2020). These applications focus on leveraging MI between successive states to encourage agents to interact with environments in various ways. Beyond exploration, MI has also been instrumental in advancing representation learning (Anand et al., 2019; Stooke et al., 2021; Nachum et al., 2019; Schwarzer et al., 2021; Mazoure et al., 2020), where it measures dependencies between states and their representations or between state-action pairs and their corresponding representations. While many of these methods utilize MI to improve an agent's performance, others apply MI in multi-agent reinforcement learning (MARL), focusing on coordination (Jaques et al., 2019; Konan et al., 2022) and promoting diversity among agents (Jiang & Lu, 2021; Li et al., 2021; Liu et al., 2022; Wang et al., 2020a).

Most existing works utilize MI to enhance the performance of a single agent or promote diversity among multiple agents in online RL. In this study, we maximize the MI between actions and policies within each state to achieve diverse behaviors in offline RL.

## 3 PRELIMINARIES

### 3.1 REINFORCEMENT LEARNING

A Markov Decision Process is defined by the tuple $M = (S, A, P, r, \rho, \gamma)$. Here, $S$ and $A$ are the state and action spaces respectively. The reward function $r(s, a)$, with state $s$ and action $a$, has a range of $[-r_{\max}, r_{\max}]$. The transition function is represented by $P(s'|s, a)$, $\rho$ is the initial state distribution, and $\gamma \in (0, 1)$ denotes the discount factor. We consider Markovian policies, $\pi \in \Pi$, which map states to distributions over actions. The value function $V^\pi(s) = \mathbb{E}_{a_t \sim \pi, s_t \sim P} \left[ \sum_{t=0}^\infty \gamma^t r(s_t, a_t) \right]$ calculates the expected discounted return from any initial state. This leads to the overall expected return:

$$\eta(\pi) = \sum_{s \in S} \rho(s) V^\pi(s).$$

The state-action value function is then defined as:

$$Q^\pi(s, a) = \mathbb{E}_{a_t \sim \pi, s_t \sim P} \left[ r(s, a) + \sum_{t=1}^\infty \gamma^t r(s_t, a_t) \mid s_0 = s, a_0 = a \right].$$

Soft Actor-Critic (SAC), a state-of-the-art off-policy actor-critic algorithm, is detailed in (Haarnoja et al., 2018). In SAC, the critic parameterized by $w$ minimizes the Bellman error as follows:

$$Q^\pi = \underset{Q^\pi}{\arg\min} \, \mathbb{E}_{(s,a,s')} \left[ r(s, a) + \gamma \left( Q_{\bar{w}}^\pi(s', \pi_\theta(a' \mid s') - \beta \log \pi_\theta(a' \mid s'))) - Q_w^\pi(s, a) \right]^2, \tag{1}$$

where $\bar{w}$ represents the target network of the critic and $\beta$ is the relative importance of the entropy term against the reward. The actor, parameterized by $\theta$, updates using the policy gradient as follows:

$$\theta \leftarrow \theta + \nabla_\theta \mathbb{E}_s \left[ Q^\pi(s, \pi_\theta(a \mid s)) - \beta \log \pi_\theta(a \mid s) \right]. \tag{2}$$

This update rule seeks to find the action extremum of the $Q$ function under the consideration of policy entropy.

## 3.2 MUTUAL INFORMATION IN DIVERSE RL SOLUTIONS

Mutual information is a fundamental concept in information theory that measures the amount of information one random variable contains about another. It is defined for two random variables, $X$ and $Y$, as follows:

$$I(X;Y) = \mathbb{E}_{p(x,y)} \left[ \log \frac{p(x,y)}{p(x)p(y)} \right] = H(X) - H(X|Y) \tag{3}$$

where $H(X)$ is the entropy of $X$, representing the uncertainty in $X$, and $H(X|Y)$ is the conditional entropy of $X$ given $Y$. This indicates that mutual information can be seen as a relative entropy that measures the reduction in the uncertainty of one random variable due to the knowledge of the other.

In Osa et al. (2022), diverse RL solutions are derived by utilizing mutual information between the trajectory random variable $\mathcal{T}$ and the policy random variable $\Pi$, which is expressed as follows:

$$I(\mathcal{T};\Pi) = H(\mathcal{T}) - H(\mathcal{T}|\Pi) = \mathbb{E}_{(\pi,\tau)\sim p(\Pi,\mathcal{T})} \left[ \log \frac{p(\tau|\pi)}{p(\tau)} \right], \tag{4}$$

where $H(\mathcal{T})$ represents the entropy of a trajectory set, indicating the variability or unpredictability of state transitions and actions over the entire state space, and $H(\mathcal{T}|\Pi)$ is the conditional entropy of the trajectory given a specific policy, reflecting the predictability of an agent's behavior when the policy is known. Throughout this paper, $p(X)$ denotes the distribution of a random variable $X$.

While Equation (4) is effective in encouraging different policies to explore varied trajectories in online RL, it presents significant implementation challenges when applied to offline RL settings. In an offline setting, policy training is inherently constrained by the dataset's transition probabilities, which are determined by the behavioral policy that generated the dataset. This constraint limits direct access to state occupation probabilities for each agent, complicating the measurement of Equation (4). To address this, we will explicitly define path diversity using Equation (4), and illustrate how our method maximizes its lower bound.

## 4 OPTIMIZING DIVERSE BEHAVIORS IN OFFLINE RL

### 4.1 DEFINITION OF DIVERSITY

In this section, we define diversity within a population of $M$ policies $\pi^1, \ldots, \pi^M$ and assume $p(\Pi)$ as a uniform distribution over these policies. We focus on two principal perspectives of diversity: path diversity and behavior diversity. Path diversity refers to the variance in the routes agents employ to achieve their goals. Behavior diversity, on the other hand, considers only the differences in the actions selected by agents, regardless of whether these actions result in different subsequent states.

#### 4.1.1 PATH DIVERSITY

We define path diversity among agents by measuring the *path uniqueness* $U_{\text{path}}^{\pi^i}$ executed by each agent $\pi^i$. To clarify this uniqueness, we first consider the dependence between the trajectory random variable and the policy random variable, quantified using mutual information:

$$I(\mathcal{T},\Pi) = \mathbb{E}_{(\pi,\tau)\sim p(\Pi,\mathcal{T})} \left[ \log \frac{p(\tau|\pi)}{p(\tau)} \right] = \mathbb{E}_{(\pi,\tau)\sim p(\Pi,\mathcal{T})} \left[ \sum_{t=0}^{T} \log \frac{p(a_t|s_t,\pi)}{p(a_t|s_t)} + \sum_{t=0}^{T} \log \frac{p^t(s_t|\pi)}{p^t(s_t)} \right],$$

where $p(\tau \mid \pi) = \prod_{t=0}^{T} p^t(s_t, a_t \mid \pi) = \prod_{t=0}^{T} p(a_t \mid s_t, \pi) p^t(s_t \mid \pi)$. This formulation represents the joint probability of state $s_t$ and action $a_t$ at time $t$ conditioned on a policy $\pi$, and $p^t(s_t \mid \pi)$ denotes the probability of being in state $s_t$ at time $t$ under that policy.

Building on mutual information, the path uniqueness $U_{\text{Path}}$ of a trajectory $\tau = (s_0, a_0, \ldots, s_T, a_T)$ executed by policy $\pi^i$ can be defined as:

$$U_{\text{Path}}^{\pi^i}(\tau) = \sum_{t=0}^{T} \log \frac{p(a_t|s_t,\pi^i)}{p(a_t|s_t)} + \sum_{t=0}^{T} \log \frac{p^t(s_t|\pi^i)}{p^t(s_t)}. \tag{5}$$

This formulation measures how likely a trajectory $\tau$ is to be executed by policy $\pi^i$. Here, the marginal state distribution at time $t$, denoted as $p^t(s_t)$, is computed by integrating over all policies, representing a mixture of state distributions across different policies:

$$p^t(s_t) = \sum_{i=1}^{M} p^t(s_t|\pi^i)p(\pi^i).$$

Additionally, $p(a_t|s_t, \pi^i)$ describes the conditional probability of taking action $a_t$ in state $s_t$ under policy $\pi^i$, and the overall probability of taking action $a_t$ in state $s_t$ across all policies, or the marginal action distribution, is given by:

$$p(a_t|s_t) = \sum_{i=1}^{M} p(a_t|s_t, \pi^i)p(\pi^i).$$

Consequently, in Equation 5, a high value for the first term indicates that only a specific agent is likely to execute the action, while a low value suggests that the action is a common choice. Similarly, a high value for the second term suggests that a specific agent predominantly visits the state, whereas a low value implies that many agents are likely to visit that state. We then define the path diversity among a collection of policies $\pi^1, \ldots, \pi^M$ and their respective trajectories $\tau^1, \ldots, \tau^M$ by aggregating the individual path uniqueness. Formally, the path diversity is given by:

$$D_{\text{Path}}^{\pi^1,\ldots,\pi^M} = \sum_{i=1}^{M} U_{\text{Path}}^{\pi^i}(\tau_i)p(\pi^i). \tag{6}$$

This sum provides a composite measure that captures the overall diversity in paths across the agents, reflecting the variability in strategies used to achieve goals within the environment.

### 4.1.2 BEHAVIOR DIVERSITY

In offline RL, while the concept of path diversity is intuitive, it is challenging to implement due to data limitations. Specifically, we only have access to conditional probabilities $p^t(s_t|\pi^B)$, where $\pi^B$ represents the behavioral policy used to generate the dataset. This restricts direct access to $p^t(s_t|\pi^i)$ for each agent, making it difficult to measure path diversity. However, we can shift our focus to behavior diversity, which measures variations in actions taken under identical states rather than differences in full trajectories.

To quantify behavior diversity, we rely on the mutual information between the action random variable and the policy random variable, expressed as $I(\mathcal{A};\Pi|\mathcal{S}) = \mathbb{E}_{(a,\pi)\sim p(\mathcal{A},\Pi)}\left[\log \frac{p(a_t|s_t,\pi)}{p(a_t|s_t)}\right]$. The behavior uniqueness $U_{\text{Behavior}}$ of a trajectory $\tau$, with respect to a policy $\pi^i$ is defined as:

$$U_{\text{Behavior}}^{\pi^i}(\tau) = \sum_{t=0}^{T} \log \frac{p(a_t|s_t, \pi^i)}{p(a_t|s_t)}$$

This metric reflects the distinctiveness of the action taken by a policy $\pi^i$ under a specific state $s_t$. Accordingly, the overall behavior diversity across a set of policies $\pi^1, \ldots, \pi^M$ and their respective trajectories $\tau^1, \ldots, \tau^M$ can be calculated by summing the behavior uniqueness for each individual policy:

$$D_{\text{Behavior}}^{\pi^1,\ldots,\pi^M} = \sum_{i=1}^{M} U_{\text{Behavior}}^{\pi^i}(\tau_i)p(\pi^i), \tag{7}$$

This approach allows us to quantify the diversity in actions taken by different policies, even when the full trajectories are not directly observable.

### 4.2 DIVERSITY OPTIMIZATION AND INTUITIONS

Let $\pi_\theta^i$ be a $\theta$-parameterized, differentiable policy. Policy gradient methods aim to optimize the expected return $\eta(\pi_\theta^i)$ by updating $\theta$ using the gradient of $\eta(\pi_\theta^i)$ with respect to $\theta$. As discussed in Section 4.1, the optimization for either path diversity or behavior diversity can be formulated as:

$$\mathcal{L}(\pi_\theta^i) = -\eta(\pi_\theta^i) - \lambda U_{\text{Path}}^{\pi_\theta^i}, \text{ or } \mathcal{L}(\pi_\theta^i) = -\eta(\pi_\theta^i) - \lambda U_{\text{Behavior}}^{\pi_\theta^i},$$

where $\lambda$ is a weight balancing performance and diversity. Although $p^t(s_t|\pi)$ is intractable in an offline setting, we can still optimize for both path and behavioral diversity. This is because $U_{\text{Behavior}}$ forms the first term of $U_{\text{Path}}$, and increasing $U_{\text{Behavior}}$ also raises the lower bound of $U_{\text{Path}}$. Specifically, the relationship between these two terms is expressed as:

$$U_{\text{Path}}^{\pi_\theta^i}(\tau) = \sum_{t=0}^{T} \log \frac{p(a_t|s_t, \pi^i)}{p(a_t|s_t)} + \sum_{t=0}^{T} \log \frac{p^t(s_t|\pi^i)}{p^t(s_t)}$$

$$\geq U_{\text{Behavior}}^{\pi_\theta^i}(\tau) = \sum_{t=0}^{T} \log \frac{p(a_t|s_t, \pi^i)}{p(a_t|s_t)}, \tag{8}$$

Accordingly, we derive a unique behavior (UB) objective function, which can be used to optimize both the quality and diversity of an RL agent:

$$\mathcal{L}(\pi_\theta^i) = -\eta(\pi_\theta^i) - \lambda U_{\text{Behavior}}^{\pi_\theta^i}. \tag{9}$$

In Figure 2, we use a 2D maze environment to illustrate how increasing behavior diversity can lead to greater path diversity. In this environment, the corridors are shown in white and the walls in gray. The goal is for agents to navigate from various starting positions to a target located at the bottom-left corner. Agents only receive rewards when they are within a 0.5-unit radius of the goal, which is visually indicated by a circle. The dataset consists of segments $(s, a, s')$, where $s$ is the current state, $a$ is the action taken, and $s'$ is the subsequent state. By maximizing behavior diversity, the experiment showed that agents followed different routes to reach the destination. This outcome is expected, as a trajectory is a sequence of states and actions, and small differences in local states accumulate over the length of the trajectory, leading to distinct paths.

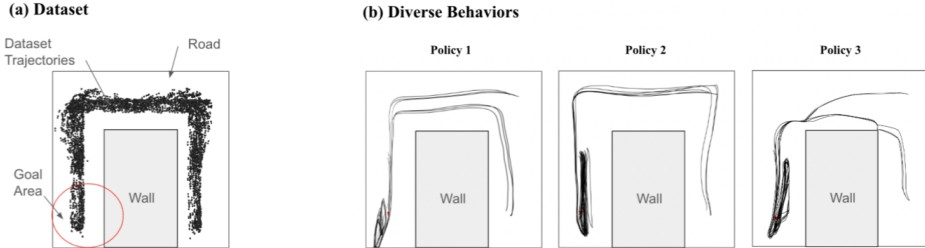

Figure 2: We demonstrate that increasing behavior diversity can lead to greater path diversity using a 2D maze environment. (a) The state-action pairs in the offline dataset. (b) **The variation in paths executed by different policies underscores the effectiveness of our approach in optimizing behavior diversity and path diversity from the same training dataset.**

### 4.3 PRACTICAL ALGORITHM

To train a policy that promotes diverse behaviors, we build our method on top of the stochastic policy optimization technique, EDAC (An et al., 2021), which is enhanced by $N$ ensemble Q-function networks. This ensemble helps reduce the overestimation of values for actions not present in the dataset, ensuring performance is maintained while pursuing diversity. The policy objective function in EDAC mirrors that of soft-actor-critic (SAC) (Haarnoja et al., 2018), and is formulated as:

$$D_{\text{KL}}\left(\pi_\theta(\cdot|s) \middle\| \frac{\exp(Q_w^\pi(s, \cdot))}{Z^\pi(s)}\right), \tag{10}$$

where $Z^\pi(s)$ normalizes the distribution. To incorporate diversity into the policy objective, we employ a reward shaping technique (Hu et al., 2020), by redefining the reward of each agent $\pi_\theta^m$ as $r'_m(s_i, a_i) = r(s_i, a_i) + \lambda U_{\text{Behavior}}^{\pi_\theta^m}(a'_{i,m}|s'_i)$, where $U_{\text{Behavior}}^{\pi_\theta^m}(a'_{i,m}|s'_i) = \log \frac{p(a'_{i,m}|s'_i, \pi_\theta^i)}{p(a'_{i,m}|s'_i)}$ represents the uniqueness of the action $a'_{i,m}$ chosen by policy $\pi_\theta^m$ at next state $s'_i$. This modification encourages policies to maximize both expected returns and behavior diversity.

---

**Algorithm 1** Offline Policy Optimaization with UB

---

1: **Input:** dataset $D = \{(s_i, a_i, r_i, s_i')\}_{i=1}^{|D|}$
2: **Initialize:** the policies $\pi_\theta^1, ..., \pi_\theta^M$, critics $Q_{w_j}^1, ..., Q_{w_j}^M$ for $j = 1, .., N$
3: **for** $t = 1$ to $T$ **do**
4:     Sample a minibatch $\{(s_i, a_i, s_i', r_i)\}_{i=1}^B$ from $D$
5:     **for** $m = 1$ to $M$ **do**
6:         Compute the target value:

$$y_i^m = r_i + \lambda U_{\text{Behavior}}^{\pi_\theta^m}(a_{i,m}'|s_i') + \gamma \min_{j=1,...,N} Q_{w_j}^m(s_i', a_{i,m}'), \text{ where } a_{i,m}' \sim \pi_\theta^m(\cdot \mid s_i')$$

7:         Update critic: for $j = 1, .., N$,    $B^{-1} \sum_{i=1}^B (y_i^m - Q_{w_j}^m(s_i, a_i))^2$
8:         Update policy:

$$B^{-1} \sum_{i=1}^B (\min_{j=1,...,N} Q_{w_j}^m(s_i, a_{i,m}) - \beta \log \pi_\theta^m(a_{i,m}|s_i)), \text{where } a_{i,m} \sim \pi_\theta^m(\cdot|s_i)$$

9:     **end for**
10: **end for**

---

Algorithm 1 provides the implementation details of our approach. Notably, while we mention multiple agents in the algorithm for clarity, in practice, *we train a single policy capable of generating multiple actions.* We adopt the same hyperparameters as the EDAC framework (An et al., 2021), with an additional hyperparameter, $\lambda$, which balances performance and diversity. This $\lambda$ was set to 1.0 across all offline datasets, except for the medium-expert Walker2d environment in the diverse D4RL dataset, where it was reduced to 0.5. Please refer to Appendix A.1 for detailed parameter setting.

## 5 RESULTS AND EVALUATION

### 5.1 COMPARISON WITH BASELINE METHODS

We conducted experiments using both the standard (Fu et al., 2020) and diversified versions[1] (Osa & Harada, 2024) of the D4RL datasets to evaluate our method. We assessed the effectiveness of our approach by comparing it with several established baseline methods designed to meet performance and diversity criteria, including DIVEOFF (Osa & Harada, 2024), CLUE (Liu et al., 2023), and SORL (Mao et al., 2024). While the baseline models generate different actions conditioned on random variables, our model outputs multiple actions simultaneously. To ensure a fair comparison, we carefully replicated the results of the baseline methods using the source codes provided by the authors [2][3][4]. Notably, as the original SORL implementation was designed for discrete action spaces, we adapted its neural network architecture to fit our continuous action space framework.

We quantified the diversity of an agent's behavior using the metric proposed in (Osa & Harada, 2024; Parker-Holder et al., 2020):

$$D_{\text{div}} = \det \left( K \left( \phi(\pi_i), \phi(\pi_j) \right)_{i,j=1}^M \right), \tag{11}$$

where $\phi(\pi) \in \mathbb{R}^l$ is the behavioral embedding of policy $\pi$, and $K : \mathbb{R}^l \times \mathbb{R}^l \to \mathbb{R}$ is a kernel function. Specifically, we used the squared-exponential kernel function:

$$k(\phi(\pi_i), \phi(\pi_j)) = \exp \left( -\frac{\|\phi(\pi_i) - \phi(\pi_j)\|^2}{2h^2} \right), \tag{12}$$

---

[1]The degree of diversity within the diversified D4RL datasets can be obtained in Appendix B.3 in (Osa & Harada, 2024).

[2]DIVEOFF: https://github.com/TakaOsa/DiveOff

[3]CLUE: https://openreview.net/forum?id=xJ7XL5Wt8iN

[4]SORL: https://github.com/cedesu/SORL/tree/main

| Datasets Type | | (a) Standard D4RL Dataset | | | | (b) Diverse D4RL Dataset | | | |
|---|---|---|---|---|---|---|---|---|---|
| Datasets | Metrics | **Ours** | DIVEOFF | CLUE | SORL | **Ours** | DIVEOFF | CLUE | SORL |
| Medium-Expert Hopper | Performance | **95.66±12.04** | 88.37±15.3 | 54.33±1.77 | 42.7±6 | **95.59±6** | 96.81±5.08 | 95.12±5.16 | 61.06±4.36 |
| | State Diversity | 0.76±0.11 | 0.31±0.3 | 0.55±0.29 | 0.98±0.02 | **0.98±0.02** | 0.1±0.08 | 0.93±0.06 | 0.95±0.03 |
| | Action Diversity | **0.46±0.22** | 0.37±0.42 | 0.37±0.21 | 0.24±0.19 | 0.66±0.26 | 0.2±0.01 | 0.09±0.08 | 0.9±0.09 |
| Medium-Expert Walker2d | Performance | **113.35±0.57** | 109±0.14 | 107.54±0.66 | 49.92±7.63 | **99.16±0.56** | 96.32±4.8 | 72.72±0.44 | 46.49±4.28 |
| | State Diversity | 0.55±0.04 | 0.15±0.11 | 0.45±0.18 | 0.92±0.06 | 0.9±0.1 | 0.6±0.37 | 0.99±0 | 0.97±0.04 |
| | Action Diversity | **0.78±0.17** | 0.15±0.14 | 0.15±0.07 | 0.52±0.18 | **0.93±0.03** | 0.25±0.33 | 0.79±0.07 | 0.89±0.07 |
| Medium-Expert Halfcheetah | Performance | **95.08±7.88** | 71.39±6.09 | 61.45±3.31 | 49.98±5.06 | 98.22±0.24 | 96.47±0.31 | 95.28±0.11 | **98.25±0.31** |
| | State Diversity | **0.82±0.34** | 0.82±0.2 | 0.64±0.37 | 0.24±0.18 | **0.98±0.03** | 0.46±0.41 | 0.71±0.26 | 0.38±0.23 |
| | Action Diversity | **0.83±0.16** | 0.43±0.35 | 0.54±0.35 | 0.38±0.11 | **0.99±0** | 0.38±0.24 | 0.74±0.14 | 0.74±0.27 |
| **Medium-Expert-Performance** | | **101.36** | 89.58 | 74.44 | 47.53 | **97.65** | 96.53 | 87.7 | 68.6 |
| **Medium-Expert-Diversity** | | **0.7** | 0.37 | 0.45 | 0.54 | **0.91** | 0.33 | 0.7 | 0.81 |
| Medium-Replay Hopper | Performance | **101.41±0.26** | 35.41±11.05 | 22.65±0.01 | 29.04±2.54 | 100.44±0.16 | **100.88±0.23** | 0.01±0.01 | 40.86±0.99 |
| | State Diversity | 0.76±0.17 | 0.4±0.21 | 0.62±0.26 | 0.84±0.12 | 0.24±0.15 | 0.02±0.03 | 0.94±0.11 | 0.77±0.07 |
| | Action Diversity | 0.42±0.25 | 0.55±0.17 | 0.86±0.18 | 0.91±0.03 | 0.27±0.14 | 0.01±0 | 0.99±0 | 0.76±0.19 |
| Medium-Replay Walker2d | Performance | **79.63±0.54** | 34.79±9.19 | 18.04±0.24 | 25±4.95 | **94.28±2.1** | 51.04±8.52 | 16.13±1.13 | 33.69±1.61 |
| | State Diversity | 0.85±0.13 | 0.49±0.37 | 0.99±0 | 0.98±0.03 | 0.52±0.15 | 0.59±0.47 | 0.99±0 | 0.95±0.04 |
| | Action Diversity | **0.97±0.03** | 0.51±0.01 | 0.9±0.15 | 0.98±0.02 | 0.72±0.14 | 0.42±0.51 | 0.99±0 | 0.92±0.04 |
| Medium-Replay Halfcheetah | Performance | **60.92±1.3** | 39.22±0.79 | 41.32±0.4 | 40.39±9.48 | **95.49±0.28** | 39.22±0.79 | 90.44±0.49 | 68.8±20.6 |
| | State Diversity | **0.94±0.03** | 0.7±0.4 | 0.07±0.04 | 0.44±0.3 | **0.99±0** | 0.48±0.44 | 0.95±0.06 | 0.16±0.06 |
| | Action Diversity | **0.98±0.02** | 0.32±0.15 | 0.12±0.01 | 0.57±0.3 | **0.99±0.01** | 0.32±0.15 | 0.98±0.02 | 0.19±0.07 |
| **Medium-Replay-Performance** | | **80.65** | 36.47 | 27.33 | 31.47 | **96.74** | 63.71 | 35.53 | 47.78 |
| **Medium-Replay-Diversity** | | **0.82** | 0.5 | 0.59 | 0.79 | 0.62 | 0.31 | **0.97** | 0.62 |
| Medium Hopper | Performance | **101.44±0.19** | 49.66±1.88 | 55.19±0.85 | 38.54±4.15 | **99.1±0.28** | 91.43±5.1 | 81.37±0.49 | 61.38±4.12 |
| | State Diversity | 0.59±0.31 | 0.14±0.1 | 0.41±0.15 | 0.82±0.23 | 0.95±0.07 | 0.19±0.28 | 0.99±0 | 0.96±0.03 |
| | Action Diversity | **0.48±0.31** | 0.33±0.1 | 0.02±0.01 | 0.33±0.24 | 0.47±0.13 | 0.34±0.45 | 0.55±0.62 | 0.9±0.07 |
| Medium Walker2d | Performance | **89.66±0.61** | 71.9±3.08 | 71.59±2.86 | 45.73±3.64 | 80.95±8.87 | **86.45±8.53** | 56.81±1.11 | 50.35±3.49 |
| | State Diversity | 0.85±0.15 | 0.86±0.1 | 0.99±0.01 | 0.96±0.03 | **0.99±0.01** | 0.68±0.36 | 0.99±0 | 0.96±0.03 |
| | Action Diversity | **0.85±0.19** | 0.22±0.23 | 0.7±0.11 | 0.51±0.26 | **0.99±0.01** | 0.59±0.36 | 0.91±0.08 | 0.85±0.23 |
| Medium Halfcheetah | Performance | **65.89±0.34** | 43.18±0.09 | 42.45±0.35 | 36.06±0.18 | 92.84±0.34 | 93.15±0.05 | 92.88±0.9 | **97.26±0.08** |
| | State Diversity | 0.72±0.21 | 0.49±0.48 | 0.73±0.11 | 0.32±0.22 | 0.98±0.01 | 0.23±0.44 | 0.75±0.2 | **0.2±0.19** |
| | Action Diversity | **0.77±0.13** | 0.14±0.01 | 0.49±0.09 | 0.15±0.08 | 0.98±0.01 | 0.45±0.3 | 0.91±0.05 | 0.27±0.23 |
| **Medium-Performance** | | **85.66** | 54.91 | 56.41 | 40.11 | **90.96** | 90.34 | 77.02 | 69.66 |
| **Medium-Diversity** | | **0.71** | 0.36 | 0.56 | 0.52 | **0.89** | 0.41 | 0.85 | 0.69 |

Table 1: The performance and diversity metrics were compared against baseline models, with results averaged across five random seeds and ten episodes per seed. For the standard D4RL dataset, we trained a policy to generate five actions (i.e., $M = 5$); for the diverse D4RL dataset, the policy output nine actions (i.e., $M = 9$). Following the methodology of (Fu et al., 2020), performance scores were normalized using $(S_o - S_r)/(S_e - S_r)$, where $S_o$, $S_r$, and $S_e$ represent the rewards achieved by the offline policy, random policy, and expert policy, respectively. The diversity scores were computed based on the method described in (Osa & Harada, 2024).

with $\phi_s(\pi_i) = \mathbb{E}_{s\sim\pi_i,P}[s]$ to evaluate state diversity and $\phi_a(\pi_i) = \mathbb{E}_{s\sim\pi_i,P}[a]$ to assess action diversity.

Table 1 presents the experimental results, where we report both the performance and diversity scores averaged over five random seeds. For clarity, the highest-performing algorithms are highlighted in **bold**, while the highest diversity scores are underlined. Additionally, the diversity scores, where the corresponding performance was within one standard deviation of the top performance models, are also highlighted in **bold** to emphasize cases where high diversity does not come at the cost of performance. As shown, our approach outperformed all baseline models in terms of overall performance and diversity across the evaluated tasks. This was true for agents trained on both the standard and diversified D4RL datasets. Notably, our method's performance remained competitive with top-performing single-agent algorithms (see Appendix A.2 and Table 5). These findings demonstrate

| Dataset | Medium-Expert | | Medium-Replay | | Medium | |
|---|---|---|---|---|---|---|
| Metrics | Performance | Diversity | Performance | Diversity | Performance | Diversity |
| **Ours** | 101.36 | **0.7** | **80.65** | **0.82** | **85.66** | **0.71** |
| **λ=0** | **104.62** | 0.34 | 80.05 | 0.41 | 85.56 | 0.28 |

Table 2: In this ablation study, setting $\lambda = 0$ means that only performance is considered during policy training. As indicated, our unique behavior (UB) objective enhances the behavior diversity of agents without sacrificing their performance.

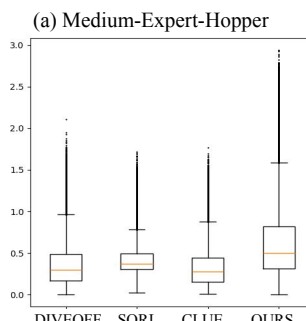 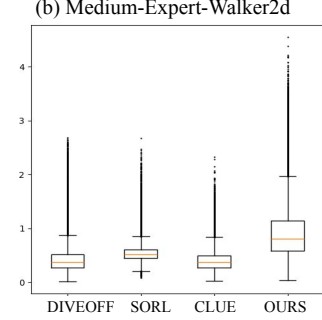 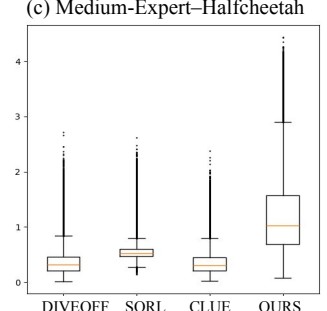

Figure 3: The Box and Whisker Plots depict the averaged distances between actions generated by policies and those sampled from the dataset under the same state. The distances were measured on the *medium-expert* level of the standard D4RL datasets. Each box represents the first quartile (Q1), the median, and the third quartile (Q3). The whiskers indicate 1.5 times the inter-quartile range (IQR), with outliers shown as individual points beyond the whiskers. **This analysis demonstrates that our method consistently selects actions from a broader range, promoting behavior diversity even in homogeneous datasets.**

that our method achieves an optimal balance between performance and diversity, offering a significant advantage in tasks that require both high efficiency and a wide range of behavioral strategies.

## 5.2 ABLATION STUDY

Our optimized diversity-and-performance algorithm for offline RL is encapsulated in the UB term (Equation 9). To assess its impact, we conducted an ablation study across nine standard D4RL datasets, comparing models trained with and without the UB term. The results, averaged over five random seeds, are presented in Table 2. A more detailed statistic for each environment can be found in Appendix A.3, Table 6. The results show that incorporating the UB objective maintains comparable performance metrics while significantly enhancing the diversity of agent behaviors. Notably, the benefit of the UB term is evident not only in the medium-replay and medium datasets, which naturally exhibit more diverse behaviors due to lower performance levels, but also in the medium-expert dataset.

## 5.3 ACTION DISTANCE FROM DATASETS

One of the key advantages of our method is its ability to learn actions outside the dataset distribution, enabling agents to exhibit diverse behaviors through intrinsic mechanisms. To quantitatively assess this benefit, we measured the average Euclidean distance between the actions selected by the agents and those typically found in the dataset. This metric is defined as follows:

$$E_{(s,a)\sim D, \hat{a}\sim\pi_\theta(\cdot|s)}[\|\hat{a} - a\|^2] \tag{13}$$

where $D$ represents the dataset and $\pi_\theta(\cdot|s)$ denotes the actions chosen by the policy. We compared our approach with DIVEOFF, SORL, and CLUE across various standard D4RL *medium-expert* datasets. Figure 3 shows the diversity of actions selected by each method. The analysis shows that our method consistently chooses actions from a broader range compared to the other

approaches. This result underscores the effectiveness of our intrinsic reward mechanism in fostering action diversity across different agents.

## 5.4 Performance in Discrete Action Environment

We further extended our evaluation to the Atari domain, where our comparison is primarily with SORL, as it is the only baseline method in our study that has also been tested on Atari environments. We executed the official SORL code available on their GitHub repository to ensure a fair comparison. For this part of our study, we set $\lambda$ to 1. We trained a model with three different policies and tested each across 10 trajectories using three random seeds to assess variability and consistency in performance. The results in Table 3 from these experiments indicate that our method not only performs well across various conditions in MuJoCo tasks but also extends effectively to other environments, including those with discrete action spaces, such as Atari.

| Atari | | Performance | State Diversity | Action Diversity |
|---|---|---|---|---|
| **SpaceInvaders** | | | | |
| | Ours | **427.6 ± 50.2** | **0.64 ± 0.29** | **0.56 ± 0.15** |
| | SORL | 422.6 ± 84.5 | 0.46 ± 0.25 | 0.27 ± 0.04 |
| **Riverraid** | | | | |
| | Ours | **1892.8 ± 309.7** | **0.57 ± 0.07** | **0.71 ± 0.16** |
| | SORL | 1751.3 ± 313.1 | 0.32 ± 0.15 | 0.50 ± 0.13 |

Table 3: Quality-diversity in discrete action environment

## 5.5 Controllable Diversity

In Wu et al. (2023), diversity is engineered through the use of user-specified Behavior Descriptors, which promote varying agent behaviors to align with different user-defined criteria. We integrate the concept from Wu et al. (2023) into our framework for scenarios where specific types of diversity are desirable. By adopting their strategy of using Behavior Descriptors, we can tailor the diversity generated by our model to fit particular user needs or to ensure alignment with targeted goals.

We conducted experiments on the Maze2d environment as depicted in Figure 2. We define $B(\pi_\theta)$ in Wu et al. (2023) as Figure 5 panel (a) in Appendix A.4, referred to as the user-specified Behavior Descriptors. Panel (b), (c), and (d) demonstrate the outcomes of this setup. Panel (b) showcases the trajectory of the target agent, which closely follows the target behavior, highlighting the efficacy of our method in embedding and controlling specific agent behaviors. In contrast, Panel (c) and (d) depict the trajectories of other agents who were not given the matching bonus but were still subject to the dataset and unique behavior rewards. These agents exhibit diverse behaviors, diverging significantly from the target, thus emphasizing the diversity achievable under our framework.

# 6 Conclusions

In this paper, we present a novel approach to enhancing diversity in offline RL environments, a field traditionally constrained by static and homogeneous training datasets. Unlike conventional methods that depend on heterogeneous datasets to cultivate diverse agent behaviors, our approach leverages mutual information to promote the development of unique and effective strategies across agents. By introducing an intrinsic reward mechanism based on the distinctiveness of an agent's actions relative to the overall action distribution, our method encourages significant diversity without compromising performance. Empirical evaluations consistently demonstrate that our framework outperforms existing methods, offering a powerful solution for generating diverse behaviors even in environments with limited data diversity. This work represents a significant advancement in offline RL, broadening the potential for deploying these strategies in more dynamic and unpredictable real-world scenarios.

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

## A  APPENDIX

### A.1  HYPERPARAMETER SETTING

For the general hyper-parameters of RL training, we follow EDAC's setting  (An et al., 2021) as shown in Table 4. For diversity optimization, the additional hyperparameter $\lambda$, which controls the weight of the uniqueness loss, is set to $1$ across all datasets, except in the *medium-expert Walker2d* environment of the diverse D4RL dataset, where it is reduced to $0.5$.

|  | Mujoco |
|---|---|
| **Optimizer** | Adam |
| **Batch Size** | 256 |
| **Learning Rate** | 0.0003 |
| **Hidden Dimension** | 256 |
| **Num of Layers** | 3 |
| **Gamma** | 0.99 |
| **Nonlinearity** | ReLU |

Table 4: Hyperparameters used in our experiments.

## A.2 COMPARISON WITH OFFLINE RL METHODS THAT LEARN A SINGLE SOLUTION

We conducted a performance comparison of leading offline reinforcement learning methods, including EDAC (An et al., 2021), Conservative Q-Learning (CQL) (Kumar et al., 2020), and Implicit Q-Learning (IQL) (Kostrikov et al., 2022), using the standard D4RL datasets. Each method was evaluated across five random seeds and ten episodes per seed. Since these methods operate within a single-policy framework, focused solely on performance, no diversity metrics are reported for them in the comparison. Notably, the performance of our diverse solutions is comparable to these single-policy methods. Detailed results from this analysis are presented in Table 5.

| Datasets | Metrics | **Ours** | **EDAC** | **CQL** | **IQL** |
|---|---|---|---|---|---|
| *Medium-Expert Hopper* | *Performance* | 95.66±12.04 | **110.7±0.1** | 96.9 ± 15.1 | 85.5 ± 29.7 |
| | *State Diversity* | **0.76±0.11** | - | - | - |
| | *Action Diversity* | **0.46±0.22** | - | - | - |
| *Medium-Expert Walker2d* | *Performance* | 113.35±0.57 | **114.7±0.9** | 109.1 ± 0.2 | 112.1 ± 0.5 |
| | *State Diversity* | **0.55±0.04** | - | - | - |
| | *Action Diversity* | **0.78±0.17** | - | - | - |
| *Medium-Expert Halfcheetah* | *Performance* | 95.08±7.88 | **106.3±1.9** | 95.0 ± 1.4 | 92.7 ± 2.8 |
| | *State Diversity* | **0.82±0.34** | - | - | - |
| | *Action Diversity* | **0.83±0.16** | - | - | - |
| *Medium-Replay Hopper* | *Performance* | **101.41±0.26** | 101.0±0.5 | 86.3 ± 7.3 | 89.6 ± 13.2 |
| | *State Diversity* | **0.76±0.17** | | | |
| | *Action Diversity* | **0.42±0.25** | | | |
| *Medium-Replay Walker2d* | *Performance* | 79.63±0.54 | **87.1±2.3** | 76.8 ± 10.0 | 75.4 ± 9.3 |
| | *State Diversity* | **0.85±0.13** | - | - | - |
| | *Action Diversity* | **0.97±0.03** | - | - | - |
| *Medium-Replay Halfcheetah* | *Performance* | **60.92±1.3** | 61.3±1.9 | 45.3 ± 0.3 | 42.1 ± 3.6 |
| | *State Diversity* | **0.94±0.03** | - | - | - |
| | *Action Diversity* | **0.98±0.02** | - | - | - |
| *Medium Hopper* | *Performance* | **101.44±0.19** | 101.6±0.6 | 61.9 ± 6.4 | 65.2 ± 4.2 |
| | *State Diversity* | **0.59±0.31** | - | - | - |
| | *Action Diversity* | **0.48±0.31** | - | - | - |
| *Medium Walker2d* | *Performance* | 89.66±0.61 | **92.5±0.8** | 79.5 ± 3.2 | 80.7 ± 3.4 |
| | *State Diversity* | **0.85±0.15** | - | - | - |
| | *Action Diversity* | **0.85±0.19** | - | - | - |
| *Medium Halfcheetah* | *Performance* | **65.89±0.34** | 65.9±0.6 | 46.9 ± 0.4 | 50.0 ± 0.2 |
| | *State Diversity* | **0.72±0.21** | - | - | - |
| | *Action Diversity* | **0.77±0.13** | - | - | - |

Table 5: We present a comparison with leading offline RL baselines, which focus exclusively on performance. Notably, our diverse solution achieves performance comparable to these single-policy methods while also offering the benefit of diverse behaviors.

## A.3 ABLATION STUDY

We conducted an ablation study across various environments in Mujoco to assess the impact of the proposed UB objective function. The detailed experimental results are provided in Table 6.

## A.4 CONTROLLABLE DIVERSITY

We conducted an controllable experiments in Maze2D to show the flexibility of our proposed UB objective function. The results are provided in Figure 5

| Datasets | Metrics | **Ours** | **λ=0** |
|---|---|---|---|
| *Medium-Expert Hopper* | *Performance* | 95.66±12.04 | **98.94±0.53** |
| | *State Diversity* | **0.76±0.11** | 0.31±0.3 |
| | *Action Diversity* | **0.46±0.22** | 0.08±0.09 |
| *Medium-Expert Walker2d* | *Performance* | **113.35±0.57** | **113.6±0.27** |
| | *State Diversity* | **0.55±0.04** | 0.46±0.28 |
| | *Action Diversity* | **0.78±0.17** | 0.48±0.08 |
| *Medium-Expert Halfcheetah* | *Performance* | 95.08±7.88 | **101.32±3.37** |
| | *State Diversity* | **0.82±0.34** | 0.36±0.29 |
| | *Action Diversity* | **0.83±0.16** | 0.37±0.13 |
| **Medium-Expert-Performance** | | 101.36 | **104.62** |
| **Medium-Expert-Diversity** | | **0.7** | 0.34 |
| *Medium-Replay Hopper* | *Performance* | **101.41±0.26** | 100.06±0.71 |
| | *State Diversity* | **0.76±0.17** | 0.07±0.06 |
| | *Action Diversity* | **0.42±0.25** | 0.47±0.28 |
| *Medium-Replay Walker2d* | *Performance* | 79.63±0.54 | **80.29±0.64** |
| | *State Diversity* | **0.85±0.13** | 0.31±0.08 |
| | *Action Diversity* | **0.97±0.03** | 0.44±0.29 |
| *Medium-Replay Halfcheetah* | *Performance* | **60.92±1.3** | 59.79±1.51 |
| | *State Diversity* | **0.94±0.03** | 0.48±0.38 |
| | *Action Diversity* | **0.98±0.02** | 0.66±0.19 |
| **Medium-Replay-Performance** | | **80.65** | 80.05 |
| **Medium-Replay-Diversity** | | **0.82** | 0.41 |
| *Medium Hopper* | *Performance* | **101.44±0.19** | **101.44±0.06** |
| | *State Diversity* | **0.59±0.31** | 0.11±0.04 |
| | *Action Diversity* | **0.48±0.31** | 0.03±0.03 |
| *Medium Walker2d* | *Performance* | 89.66±0.61 | **90.47±0.75** |
| | *State Diversity* | **0.85±0.15** | 0.73±0.13 |
| | *Action Diversity* | **0.85±0.19** | 0.1±0.04 |
| *Medium Halfcheetah* | *Performance* | **65.89±0.34** | 64.78±0.36 |
| | *State Diversity* | **0.72±0.21** | 0.22±0.03 |

Table 6: Detailed Results of the ablation study. In this table, setting $\lambda = 0$ means that only performance is considered during policy training. As indicated, our unique behavior (UB) objective enhances the behavior diversity of agents without sacrificing their performance.

## A.5 DERIVATION DETAILS IN SECTION 4

In our unique behavior approach, we focus exclusively on computing the value $p(a_t|s_t)$, which is directly determined from the available policy outputs. Importantly, our method does not require the estimation of $p_t(s_t)$, the state transition probability, which simplifies the computational process and reduces the potential for introducing estimation errors.

Specifically, In the formula $p(a_t|s_t) = \sum_{i=1}^{M} p(a_t|s_t, \pi^i)p(\pi^i)$ the term $p(a_t|s_t, \pi^i)$ is defined as $\pi^i(a_t|s_t)$, where $\pi^i$ represents the policy model's output for policy $i$. Without loss of generality, we assume a uniform distribution over the policies where $p(\pi^i) = \frac{1}{M}$, for each $i$. As we have full knowledge of each policy $\pi^i(a_t|s_t)$, there is no need to estimate these probabilities. This direct

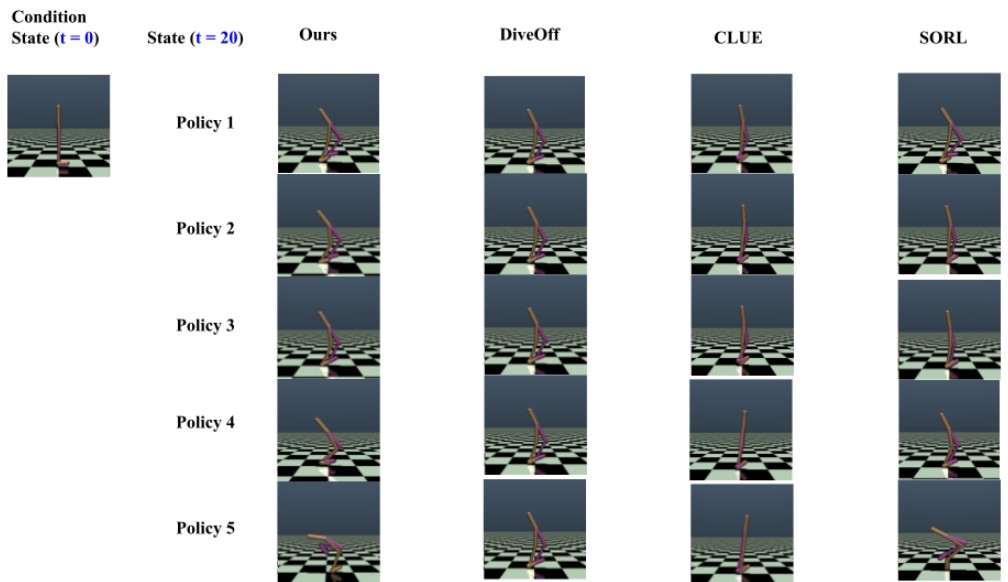

Figure 4: To visually compare the diversity between policies trained with baseline methods and our approach, we initialized the policies from a consistent initial state at $t = 0$ and rendered the resulting states after 20 steps.

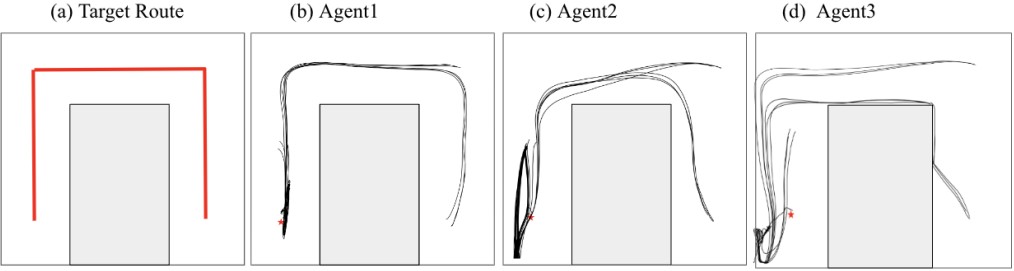

Figure 5: Controllable Diversity

utilization avoids the introduction of additional bias or variance that often accompanies estimation processes.

In Inequation 8, we adapt the mutual information concept from Equation 4 to the context between the state $\mathcal{S}$ and the policy $\Pi$, we have: $I(\mathcal{S}; \Pi) = H(\mathcal{S}) - H(\mathcal{S}|\Pi) = \mathbb{E}_{(\pi,s) \sim p(\Pi, \mathcal{S})} \left[ \log \frac{p(s|\pi)}{p(s)} \right]$ Given the non-negative nature of mutual information, this formulation supports the validity of Inequation 8.

