# OpenReview forum: "Learn out of the box: optimizing both diversity and performance in Offline Reinforcement Learning"
_ICLR.cc/2025/Conference — Submitted to ICLR 2025_

### Official Review · Reviewer_XcFd · 2024-10-19

**Soundness:** 2
**Presentation:** 3
**Contribution:** 2
**Rating:** 6
**Confidence:** 4

**Summary:**

This paper introduces a novel approach to offline reinforcement learning that enhances behavioral diversity while maintaining performance. Unlike traditional methods that depend on diverse datasets, it maximizes mutual information between actions and policies, enabling agents to learn varied behaviors even from homogeneous data. The approach incorporates an intrinsic reward mechanism and the Ensemble-Diversified Actor-Critic framework to handle out-of-distribution risks. Experimental results on D4RL benchmarks show superior performance and diversity compared to existing methods, making it a promising solution for more robust offline RL applications.

**Strengths:**

1. Introducing a novel method to enhance behavioral diversity in offline reinforcement learning, addressing limitations of traditional approaches that rely on dataset heterogeneity.
2. Leveraging mutual information, the method enables agents to learn a range of behaviors beyond those present in the dataset.
3. Integrating the EDAC framework helps manage out-of-distribution risks, ensuring that the diversity achieved does not come at the cost of performance.

**Weaknesses:**

1. The paper lacks clear explanations of some key concepts, leading to ambiguity in how the proposed approach connects to its claims, such as the relevance of the title and certain methodological terms.
2. There are insufficient details about the implementation and underlying mechanisms of the method. This makes it challenging for readers to fully understand or replicate the approach.
3. The comparisons with existing methods are limited, particularly in demonstrating the proposed method's advantages over other high-performance algorithms. Broader benchmarks and more comprehensive comparisons would strengthen the claims.
4. Some theoretical aspects are not well-justified, leading to questions about the validity and robustness of the approach. Clearer explanations and references to prior work are needed.

#### If these issues are adequately addressed, which are detailed in the following questions, the paper could offer a valuable contribution to the field, particularly in the area of improving diversity in offline reinforcement learning.

**Questions:**

1. The phrase 'out of the box' in the title is not clearly connected to specific concepts or points within the paper. The authors should clarify what 'box' refers to in the context of their method.
2. Regarding the abstract, after reviewing the full paper, I found that the comparison between heterogeneous and less diverse datasets is not explicitly addressed in the methodology or experiments. The claimed effectiveness of the approach across different data types has not been adequately substantiated. Could the authors specify which datasets are considered heterogeneous and which are less diverse, and clarify this distinction in the methodology or experiment sections?
3. How important is diversity for reinforcement learning, particularly in offline RL? What are the main benefits of promoting diversity in this context? The authors should elaborate more on the significance and implications of this aspect in their research.
4. What distinguishes the current mutual information-based approach from other methods, including the baseline approaches like DIVEOFF, CLUE, and SORL? The authors mention that DIVEOFF also maximizes mutual information, so why does their method achieve better diversity while balancing performance? What is the underlying reason for this difference?
5. Regarding Figure 1, the comparison between the proposed method and SORL does not show a significant difference. In my view, their diversity levels appear quite similar. Could the authors provide another example where the distinction between the methods is more pronounced?
6. Why does the approach adopt an ensemble method (EDAC) to avoid over-estimation? How does this compare to other techniques like conservative Q-learning or conservative policy gradient methods? What are the advantages of using EDAC over these alternatives?
7. The paper states, "However, a challenge arises when the dataset is homogeneous, or when distant latent vectors are decoded into similar actions." What exactly does it mean for distant vectors to be decoded into similar actions? Could the authors provide a more detailed explanation of this concept?
8. The paper states, "Agents are rewarded for behaving differently from each other during training, providing direct guidance that allows policies to learn behaviors even beyond the dataset distribution." Why does the proposed method enable learning behaviors beyond the dataset distribution? How does this approach differ from importance sampling-based methods, and why did the authors choose not to use importance sampling?
9. The estimation formula $p\left(a_t \mid s_t\right)=\sum_{i=1}^M p\left(a_t \mid s_t, \pi^i\right) p\left(\pi^i\right)$ on page 5 is intriguing, but can it truly provide an accurate estimate of $p\left(a_t \mid s_t\right)$? Are there any potential issues with bias or high variance in this approach? The same concern applies to the formula $p^t\left(s_t\right)=\sum_{i=1}^M p^t\left(s_t \mid \pi^i\right) p\left(\pi^i\right)$. Could the authors explain the rationale behind using these estimations? If this method is derived from existing research, it would be helpful to cite those sources and provide further clarification.
10. In Inequation (8), is the second term of $U_{\text{Path}}^{\pi_{\theta}^i}(\tau) = \sum_{t=0}^{T} \log \left( \frac{p(a_t \mid s_t, \pi^i)}{p(a_t \mid s_t)} \right) + \sum_{t=0}^{T} \log \left( \frac{p^t(s_t \mid \pi^i)}{p^t(s_t)} \right)$ always positive? It is not clear to me why this inequation holds. Could the authors provide further explanation?
11. To which part of the paper does Figure 2 correspond? The authors did not clearly indicate this connection.
12. In Algorithm 1, how is the policy update in line 8 implemented? Did you use the policy gradient theorem? The authors should provide more details on how the policies are updated.
13. The notation $\phi(\pi_i) = E_{s \sim \pi_i, P}[s]$ is used to evaluate state diversity, while $\phi(\pi_i) = E_{s \sim \pi_i, P}[a]$ is used to assess action diversity. However, it appears that the same notation $\phi(\pi_i)$ is being used for two different values. Could the authors clarify this?
14. In Table 4, the performance of the EDAC method appears to be better than that of the proposed approach. Under what conditions should we choose the proposed method for its diversity benefits, despite the lower performance compared to EDAC? Is there a standard metric or conventional score to evaluate this trade-off between diversity and performance?
15. I noticed that the proposed method appears to outperform DIVEOFF, SORL, and CLUE in both performance and diversity. Could the authors also evaluate the diversity of other high-performance algorithms, such as EDAC, CQL, and IQL? Furthermore, are the observed differences in actions genuinely due to enhanced diversity, or could they be attributed to insufficient training of the policies (e.g., SORL), which may have been limited to a single, suboptimal policy?
16. Is this method applicable only to offline learning? If not, please consider discussing its potential for online scenarios in the future work section.

---

> ### Author Response · Authors · 2024-11-20
> **Author Response**
>
> We thank reviewer XcFd for the insightful comments. Below, we address the questions raised by the reviewer. We hope the replies could help the reviewer further recognize our contributions. Thank you.
>
> **Q1: The phrase 'out of the box' in the title is not clearly connected to specific concepts or points within the paper. The authors should clarify what 'box' refers to in the context of their method.**
>
> A1: The phrase "out of the box" metaphorically refers to the capability to transcend the typical constraints posed by the training dataset in offline reinforcement learning. In the conventional setup, RL agents are often strictly to learn actions within the dataset. Our method, however, is designed to enable learning behaviors outside the dataset. The box implies the data constraint. We are happy to rephrase the title if the phrase "out of the box" is not explicit.
>
>
> **Q2: The paper states, "Agents are rewarded for behaving differently from each other during training, providing direct guidance that allows policies to learn behaviors even beyond the dataset distribution." Why does the proposed method enable learning behaviors beyond the dataset distribution? How does this approach differ from importance sampling-based methods, and why did the authors choose not to use importance sampling?**
>
> A2: The key to enabling behaviors beyond the dataset distribution lies in the design, where policies are rewarded for behaving differently from each other during training. Offline RL is not a mere behavior cloning; rather, policies are guided through value functions that assess and update model parameters based on the policy's actions. Under offline constraints, the value function is trained solely on pre-collected data. However, when policies are encouraged to maintain distinct action distributions, they may naturally be pushed beyond the dataset's scope. This approach fundamentally contrasts with importance sampling, which continues to sample from the original dataset distribution and does not facilitate policy actions outside of the dataset range.
>
> **Q3. What distinguishes the current mutual information-based approach from other methods, including the baseline approaches like DIVEOFF, CLUE, and SORL? The authors mention that DIVEOFF also maximizes mutual information, so why does their method achieve better diversity while balancing performance? What is the underlying reason for this difference?**
>
> A3: Our approach and DIVEOFF differ fundamentally in the types of actions they focus on for mutual information (MI) maximization, leading to distinct outcomes in terms of policy behavior and diversity.
> Our method maximizes MI between the policy and the actions it selects during training. These actions are generated by the policy itself, which means they are not restricted to existing state-action pairs within the dataset. The only constraint is that different policies should exhibit noticeable behavioral diversity, thereby encouraging exploration even beyond the dataset distribution.
> In contrast, DIVEOFF, along with other baseline methods like CLUE and SORL, focuses on actions within the dataset, specifically the state-action pairs pre-collected in the training data. DIVEOFF maximizes MI between latent vectors and these state-action pairs, inherently limiting the policy’s behavior to what the dataset already contains.
> By emphasizing MI in the policy’s own selected actions, our approach achieves diversity beyond the dataset. EDAC further ensures that this exploration is done with reliable value estimates, enabling the policy to balance performance and diversity effectively.

---

> ### Author Response · Authors · 2024-11-20
> **Author Response**
>
> **Q4. Why does the approach adopt an ensemble method (EDAC) to avoid over-estimation? How does this compare to other techniques like conservative Q-learning or conservative policy gradient methods? What are the advantages of using EDAC over these alternatives?**
>
> A4: In offline RL, mitigating action value over-estimation is crucial, with two primary strategies often applied: conservative methods and uncertainty awareness.
>
> **Conservative Methods:** Techniques like Conservative Q-Learning (CQL) and TD3+BC focus on guiding the agent to favor actions within the dataset, thereby minimizing over-estimated value predictions and stabilizing learned policies. However, the conservative approach limits the exploration of novel actions outside the training data, which may hinder objectives related to diversity and exploratory behaviors.
>
> **Uncertainty Awareness with EDAC:** Methods such as Ensemble Distributional Actor-Critic (EDAC) use ensemble models to quantify uncertainty in value estimations. When the agent encounters actions outside the dataset, the ensemble provides multiple value estimates, enabling a more robust average and variance calculation. This ensemble-based approach allows the policy to better assess the potential risks and rewards associated with OOD actions, providing more reliable estimations even when extrapolating beyond the dataset. By focusing on actions with lower uncertainty, EDAC naturally enables policies to explore a broader action space while controlling for over-estimation, aligning with our goal of fostering diversity without compromising performance.
>
> **Q5. In Algorithm 1, how is the policy update in line 8 implemented? Did you use the policy gradient theorem? The authors should provide more details on how the policies are updated.**
>
> A5: In Algorithm 1, the policy update mechanism employed in line 8 is based on the framework used in Soft Actor-Critic (SAC), with a key modification tailored to enhance robustness and stability. Specifically, while SAC typically updates policies using a standard policy gradient method, we select the Q-function that provides the minimum value for the current policy from among those available in an ensemble of Q-functions (EDAC). This selection criterion is promoting more stable learning by mitigating the risk of overestimation in offline RL, particularly when policy's actions are outside the dataset.
>
> **Q6. The paper states, "However, a challenge arises when the dataset is homogeneous, or when distant latent vectors are decoded into similar actions." What exactly does it mean for distant vectors to be decoded into similar actions? Could the authors provide a more detailed explanation of this concept?**
>
> A6: A decoder taking distinct inputs is not guaranteed to generate different outcomes. This issue is analogous to mode collapse in generative models, where diverse inputs (latent vectors) lead to indistinguishable outputs, failing to capture the potential diversity the model is theoretically capable of expressing. Latent conditioned policies, like those employed in DIVEOFF, map latent vectors to actions within a dataset, and likely suffer from the same problem. This phenomenon is evident in the results for DIVEOFF, as depicted in Figure 1, where different latent vectors yield similar policy behaviors.
>
> **Q7. Regarding the abstract, after reviewing the full paper, I found that the comparison between heterogeneous and less diverse datasets is not explicitly addressed in the methodology or experiments. The claimed effectiveness of the approach across different data types has not been adequately substantiated. Could the authors specify which datasets are considered heterogeneous and which are less diverse, and clarify this distinction in the methodology or experiment sections?**
>
> A7: Standard D4RL Dataset is generated using a Soft Actor-Critic (SAC) policy, which typically results in homogeneous behavior patterns. In contrast, the Diverse D4RL Dataset is collected using a latent-conditioned policy that is specifically designed to capture a wider range of behaviors, making it more heterogeneous. The Diverse D4RL Dataset exhibits greater diversity compared to the Standard D4RL Dataset due to the inclusion of multiple latent-conditioned policies. We apologize for not providing a detailed description of the Diverse D4RL Dataset earlier, as the dataset was collected by the authors of SORL. We will revise the manuscript to more clearly distinguish between the Standard and Diverse D4RL datasets, and to clarify their differences in both the methodology and experiments sections.

---

> ### Author Response · Authors · 2024-11-20
> **Author Response**
>
> **Q8. How important is diversity for reinforcement learning, particularly in offline RL? What are the main benefits of promoting diversity in this context? The authors should elaborate more on the significance and implications of this aspect in their research.**
>
> A8: Below, we discuss the advantages of diversity for RL in different aspects.
>
> **Novelty and Engagement:** In applications such as conversational agents, diversity ensures that interactions remain engaging and novel, even across many sessions with the same user. This can prevent conversational stagnation and maintain user interest and satisfaction over time.
>
> **Preference and Personalization:** Diversity allows for the development of personalized policies that can cater to specific preferences or requirements. In scenarios like recommendation systems or adaptive interfaces, where user satisfaction is paramount, having a diverse array of strategies enables the system to tailor its responses more effectively to individual users.
>
> **Simulations:** In simulations involving groups of agents, such as in crowd simulation or multi-agent systems, diversity can lead to more realistic and high-fidelity models of group behavior.
>
> From the perspective of applications, diversity is equally important in online and offline RL since they differ mainly in training environments.
>
> **Q9. In Table 4, the performance of the EDAC method appears to be better than that of the proposed approach. Under what conditions should we choose the proposed method for its diversity benefits, despite the lower performance compared to EDAC? Is there a standard metric or conventional score to evaluate this trade-off between diversity and performance?**
>
> A9: **More than performance:** While the EDAC method may exhibit better performance on certain benchmarks, our proposed method offers unique advantages in scenarios where diversity in agent behavior is critical. This is particularly important in applications requiring human-agent interaction or multi-agent environments. For example, in interactive applications such as gaming or conversational agents, diverse responses or actions can enhance user engagement, ensuring sustained interest and participation over time. In such contexts, the benefits of diversity outweigh the marginal differences in performance.
>
> **Evaluating Trade-offs:** As for evaluating the trade-off between diversity and performance, we acknowledge that there is currently no widely accepted standard metric for this purpose. However, we address this trade-off in our analysis (Figure 1), which visually represents the relationship between these factors. Ideally, the most desirable policies are those situated in the upper-right quadrant, demonstrating both high diversity and strong performance.
>
> **Q10. I noticed that the proposed method appears to outperform DIVEOFF, SORL, and CLUE in both performance and diversity. Could the authors also evaluate the diversity of other high-performance algorithms, such as EDAC, CQL, and IQL? Furthermore, are the observed differences in actions genuinely due to enhanced diversity, or could they be attributed to insufficient training of the policies (e.g., SORL), which may have been limited to a single, suboptimal policy?**
>
>
> A10: Algorithms like EDAC, CQL, and IQL typically train a single policy. For a given input state $s$, the action produced by these policies remains identical regardless of how many times they encounter $s$, resulting in a diversity score of zero. For evaluation, DIVEOFF’s authors modify IQL by integrating it with a Variational Autoencoder (VAE) to create a latent-conditioned version (IQL+VAE). As reported in their study, DIVEOFF achieves higher diversity compared to IQL+VAE. Our method further surpasses DIVEOFF, demonstrating superior performance and diversity.
>
> For policies trained using EDAC, CQL, and IQL, insufficient training would result in poor decisions overall but would not affect the deterministic nature of their actions for a given state $s$. In contrast, methods such as SORL, DIVEOFF, CLUE, and our proposed approach produce varying behaviors for the same state $s$ due to the use of latent vectors or policy indices. These differences in behavior are intrinsic to the model's design and cannot be attributed to insufficient training. Instead, they reflect the intended diversity mechanisms embedded within these methods.

---

> ### Author Response · Authors · 2024-11-20
> **Author Response**
>
> **Q11. The estimation formula $p(a_t | s_t) = \sum_{i=1}^M p(a_t | s_t, \pi^i) p(\pi^i)$ on page 5 is intriguing, but can it truly provide an accurate estimate of $p(a_t∣s_t)$? Are there any potential issues with bias or high variance in this approach? The same concern applies to the formula $p^t(s_t) = \sum_{i=1}^M p^t(s_t | \pi^i) p(\pi^i)$. Could the authors explain the rationale behind using these estimations? If this method is derived from existing research, it would be helpful to cite those sources and provide further clarification.**
>
> A11:
> In our unique behavior approach, we focus exclusively on computing the value $p(a_t | s_t)$, which is directly determined from the available policy outputs. Importantly, our method does not require the estimation of $p_t(s_t)$, the state transition probability, which simplifies the computational process and reduces the potential for introducing estimation errors.
>
> Specifically, In the formula $p(a_t | s_t) = \sum_{i=1}^{M} p(a_t | s_t, \pi^i) p(\pi^i) $ the term $p(a_t | s_t, \pi^i)$ is defined as $\pi^i(a_t | s_t) $, where $ \pi^i $ represents the policy model's output for policy $ i $. Without loss of generality, we assume a uniform distribution over the policies where $ p(\pi^i) = \frac{1}{M} $, for each $ i $. As we have full knowledge of each policy $\pi^i(a_t | s_t) $, there is no need to estimate these probabilities. This direct utilization avoids the introduction of additional bias or variance that often accompanies estimation processes.
>
> **Q12. In Inequation (8), is the second term of $U_{\text{Path}}^{\pi^i}(\tau) = \sum_{t=0}^{T} \log \left( \frac{p(a_t | s_t, \pi^i)}{p(a_t | s_t)} \right) + \sum_{t=0}^{T} \log \left( \frac{p^t(s_t | \pi^i)}{p^t(s_t)} \right)$ always positive? It is not clear to me why this inequation holds. Could the authors provide further explanation?**
>
> A12: Adapting the mutual information concept from Equation 4 to the context between the state $ \mathcal{S}$ and the policy $\Pi$, we have: $ I(\mathcal{S}; \Pi) = H(\mathcal{S}) - H(\mathcal{S}|\Pi) = \mathbb{E}_{(\pi, s) \sim p(\Pi, \mathcal{S})} \left[ \log \frac{p(s|\pi)}{p(s)} \right] $ Given the non-negative nature of mutual information, this formulation supports the validity of Inequation (8).
>
>
> **Q13. To which part of the paper does Figure 2 correspond? The authors did not clearly indicate this connection.**
>
> A13: Thank you for pointing out the omission. Figure 2 corresponds to the discussion on lines 284 to 293 of the paper. We have revised the document as: In Figure 2, we use a 2D maze environment to illustrate how increasing behavior diversity can lead to greater path diversity. In this environment, the corridors are shown in white and the walls in gray. The goal is for agents to navigate from various starting positions to a target located at the bottom-left corner. Agents only receive rewards when they are within a 0.5-unit radius of the goal, which is visually indicated by a circle. The dataset consists of segments $(s, a, s')$, where $s$ is the current state, $a$ is the action taken, and $s'$ is the subsequent state. By maximizing behavior diversity, the experiment showed that agents followed different routes to reach the destination. This outcome is expected, as a trajectory is a sequence of states and actions, and small differences in local states accumulate over the length of the trajectory, leading to distinct paths.
>
> **Q14. The notation  $\phi(\pi_i) = E_{s \sim \pi_i, P}[s] $ is used to evaluate state diversity, while $\phi(\pi_i) =E_{s \sim \pi_i, P} [a]$ is used to assess action diversity. However, it appears that the same notation $\phi(\pi_i)$  is being used for two different values. Could the authors clarify this?**
>
> A14: In our formulation, $\phi(\pi_i)$ serves as a generic notation for a behavioral embedding function within the kernel calculation. This function is adapted based on the context—either state or action diversity. To alleviate any confusion and improve the clarity of our methodology, we redefine the notation to reflect these distinct applications more clearly: $\phi_s(\pi_i) = E_{s \sim \pi_i, P} [s]$ to evaluate state diversity and $\phi_a(\pi_i) = E_{s \sim \pi_i, P} [a]$ to assess action diversity. We have revised the document and have uploaded the new PDF version. Please refer to this latest version for the updated content.

---

> ### Author Response · Authors · 2024-11-20
> **Author Response**
>
> **Q15. Is this method applicable only to offline learning? If not, please consider discussing its potential for online scenarios in the future work section.**
>
> A15: While our current implementation and evaluations primarily focus on offline learning scenarios, the underlying principles of our method have broader applicability, including potential extensions to online learning environments. The core principle of our approach—encouraging unique behavior among agents—remains effective regardless of the learning context. This is because the incentive for agents to adopt diverse and unique actions is not inherently linked to the data being static or dynamically generated. In online learning, where agents interact with an environment and decisions are immediately reflected in the environment, the same intrinsic rewards designed to promote behavioral diversity can be employed. These rewards would guide the agent to explore novel actions during its interaction with the environment, enhancing its ability to adapt to evolving situations and potentially improving its exploration strategies.

---

> > ### Comment · Reviewer_XcFd · 2024-11-24
> > **Response to authors**
> >
> > Thank you for addressing my previous questions. I appreciate the efforts the authors have invested in this work. However, I still have significant concerns regarding the core contributions. Specifically, the diversity aspect of offline reinforcement learning has not been clearly articulated. As noted by other reviewers, the key distinction between the mutual information approach in this paper and its application in existing works requires further clarification.
> >
> > Additionally, the concept of unique behavior (UB) would benefit from more rigorous theoretical derivations—potentially included as supplementary material in the Appendix—to demonstrate its advantages. I also recommend conducting additional ablation studies to provide stronger empirical evidence for its effectiveness.
> >
> > Therefore, while the paper shows good potential, it needs a more explicit declaration of its contributions and further validation of their effectiveness. In light of these considerations, I have increased my score to reflect my optimism for the authors' future revisions and improvements.

---

> > > ### Author Response · Authors · 2024-11-25
> > > **Author Response**
> > >
> > > We appreciate the insightful feedback provided by Reviewer XcFd. To address the concerns raised, we have organized our response into two main sections: 1. core contribution and 2. Theoretical derivations and ablation study of unique behavior (UB). **Should there be any confusion or further questions regarding our methods or results, we are more than willing to provide additional clarification.**
> > >
> > > # 1. Core contribution:
> > >
> > > ## 1.1 Key distinction in methods
> > >
> > > The key distinction between our method and others such as DIVEOFF, CLUE, and SORL **does not lie merely in the use of mutual information, but also in how they use offline data**. Specifically, these baseline methods use Expectation Maximization  to cluster offline data and encourage policies to learn behaviors from respective clusters. In contrast, our approach supplements the learning from offline data with an additional unique behavior (UB) reward, which demands policies to exhibit distinct behaviors from one another. This UB reward and the offline RL learning conflict partially. A significant weight on the UB reward leads to greater behavioral divergence among the policies, although this may cause them to deviate substantially from the dataset. This is how our method can encourage learning of behaviors not present in the dataset, a capability not shared by DIVEOFF, CLUE, and SORL. As depicted in Figure 3, this differentiation is clear.
> > >
> > > ## 1.2 Key advantage: diversity-quality trade-off
> > > **Our method is the only method that achieves both diversity and performance simultaneously.** As demonstrated in Figure 1, other methods are positioned either in the bottom right or top left corners, indicating a trade-off between diversity and performance. Addressing this diversity-quality trade-off is a complex challenge, which we further elucidate by summarizing the results from Table 1 into a concise table within our paper. This summary shows that while the diversity of the dataset significantly influences the behavioral diversity of baseline methods, it has minimal impact on the performance of our proposed method.
> > >
> > > | Datasets                   | Metrics               | | Ours   | DIVEOFF | CLUE  | SORL | | Ours   | DIVEOFF | CLUE  | SORL  |
> > > |----------------------------|--------------------|----|--------|---------|-------|-----|--|--------|---------|-------|-------|
> > > |                            |                        | **Standard D4RL Dataset**| |       |       |       | **Diverse D4RL Dataset** | |       |       |       |
> > > | **Medium-Expert** |  Performance                    |  | 101.36 | 89.58   | 74.44 | 47.53| | 97.65  | 96.53   | 87.7  | 68.6  |
> > > | **Medium-Expert**    |    Diversity                 |   | 0.7    | 0.37    | 0.45  | 0.54 | | 0.91   | 0.33    | 0.7   | 0.81  |
> > > | **Medium**         | Performance                     |  | 85.66  | 54.91   | 56.41 | 40.11| | 90.96  | 90.34   | 77.02 | 69.66 |
> > > | **Medium**           | Diversity                      | | 0.71   | 0.36    | 0.56  | 0.52 | | 0.89   | 0.41    | 0.85  | 0.69  |
> > >
> > > ## 1.3 Robustness
> > >
> > > In addition to **the 18 distinct experimental conditions** discussed in our main paper, **we have extended our evaluation to the Atari domain for this rebuttal.** Here, our primary comparison is with SORL, the only baseline method from our study that has also been applied to Atari environments. We executed the official SORL code available on their GitHub repository to ensure a fair comparison. For this part of our study, we set $\lambda$( the weight for  $U_{\text{Behavior}}$) to 1. We trained a model with three different policies and tested each across 10 trajectories using three random seeds to assess variability and consistency in performance. The results from these experiments indicate that our method not only performs well across various conditions in MuJoCo tasks but also extends effectively to other environments, including those with discrete action spaces, such as Atari.
> > >
> > > | Atari          | Performance         | State Diversity     | Action Diversity   |
> > > |----------------|---------------------|---------------------|--------------------|
> > > | **SpaceInvaders** |                     |                     |                    |
> > > | Ours           | **427.6 ± 50.2**        | **0.64 ± 0.29**        | **0.56 ± 0.15**        |
> > > | SORL           | 422.6 ± 84.5        | 0.46 ± 0.25         | 0.27 ± 0.04        |
> > > | **Riverraid**     |                     |                     |                    |
> > > | Ours           | **1892.8 ± 309.7**      | **0.57 ± 0.07**         | **0.71 ± 0.16**        |
> > > | SORL           | 1751.3 ± 313.1      | 0.32 ± 0.15         | 0.50 ± 0.13        |

---

> > > > ### Author Response · Authors · 2024-11-25
> > > > **Author Response**
> > > >
> > > > # 2. Theoretical derivations and ablation study of unique behavior (UB):
> > > >
> > > > ## 2.1 Theoretical derivation
> > > > **Below is our theoretical derivation. We will add this paragraph to our appendix:**
> > > >
> > > > In our unique behavior approach, we focus exclusively on computing the value $p(a_t | s_t)$, which is directly determined from the available policy outputs. Importantly, our method does not require the estimation of $p_t(s_t)$, the state transition probability, which simplifies the computational process and reduces the potential for introducing estimation errors.
> > > >
> > > > Specifically, In the formula $p(a_t | s_t) = \sum_{i=1}^{M} p(a_t | s_t, \pi^i) p(\pi^i)$ the term $p(a_t | s_t, \pi^i)$ is defined as
> > > >  $\pi^i(a_t | s_t)$, where $\pi^i$ represents the policy model's output for policy $i$. Without loss of generality, we assume a uniform distribution over the policies where $p(\pi^i) = \frac{1}{M}$, for each $i$. As we have full knowledge of each policy $\pi^i(a_t | s_t)$, there is no need to estimate these probabilities. This direct utilization avoids the introduction of additional bias or variance that often accompanies estimation processes.
> > > >
> > > > In equation (8), we adapt the mutual information concept from Equation 4 to the context between the state $\mathcal{S}$ and the policy $\Pi$, we have: $I(\mathcal{S}; \Pi) = H(\mathcal{S}) - H(\mathcal{S}|\Pi) = \mathbb{E}_{(\pi, s) \sim p(\Pi, \mathcal{S})} \left[ \log \frac{p(s|\pi)}{p(s)} \right]$. Given the non-negative nature of mutual information, this formulation supports the validity of Inequation (8).
> > > >
> > > > ## 2.2 Ablation study
> > > >
> > > > **In Table 2 and Appendix A.3 of our manuscript, we present an ablation study of the Unique Behavior component to demonstrate its impact.** Compared to a naive multi-model policy approach, our method significantly enhances diversity across various conditions. Specifically, diversity scores in the medium-expert, medium-replay, and medium categories have increased to 0.7, 0.82, and 0.71, respectively, up from 0.34, 0.41, and 0.28.
> > > >
> > > > **Moreover, in this rebuttal, we further demonstrate the flexibility of our method.** In [Wu S et al., 2023], diversity is engineered through the use of user-specified Behavior Descriptors, which promote varying agent behaviors to align with different user-defined criteria. We conducted experiments (https://imgur.com/a/e0TKFsS ) on the Maze2d environment as depicted in Figure 2 of our main paper to demonstrate our method’s flexibility.
> > > >
> > > > We define $B(\pi_θ)$ in  [Wu S et al., 2023]  as panel (a), referred to as the user-specified Behavior Descriptors. Panel (b), (c), and (d) demonstrate the outcomes of this setup. Panel (b) showcases the trajectory of the target agent, which closely follows the target behavior, highlighting the efficacy of our method in embedding and controlling specific agent behaviors. In contrast, Panel (c) and (d) depict the trajectories of other agents who were not given the matching bonus but were still subject to the dataset and unique behavior rewards. These agents exhibit diverse behaviors, diverging significantly from the target, thus emphasizing the diversity achievable under our framework.
> > > >
> > > > [Wu S et al., 2023] Wu S et al. Quality-similar diversity via population based reinforcement learning[C]//The Eleventh International Conference on Learning Representations. 2023.

---

> > > > > ### Comment · Reviewer_XcFd · 2024-11-26
> > > > > **Response to authors**
> > > > >
> > > > > Thank you for addressing my questions in further detail. I sincerely appreciate the efforts the authors have put into this work, which indeed demonstrates notable improvements over previous approaches and offers unique advantages. I have adjusted my score upward to reflect my recognition of the authors' valuable contributions.
> > > > >
> > > > > That said, I believe the current approach of simply incorporating the unique behavior (UB) reward may appear somewhat straightforward, and its effectiveness would benefit from stronger theoretical guarantees or deeper insights. I encourage the authors to consider exploring this aspect further in their future work to solidify and enhance the impact of their contributions.

---

### Official Review · Reviewer_BSoj · 2024-11-02

**Soundness:** 3
**Presentation:** 3
**Contribution:** 2
**Rating:** 5
**Confidence:** 3

**Summary:**

The paper introduces a novel approach to enhance behavioral diversity in offline reinforcement learning without compromising performance, leveraging mutual information to encourage diverse strategies among agents. It employs an Ensemble-Diversified Actor-Critic method to mitigate the risks associated with out-of-distribution actions, ensuring quality alongside diversity. The innovation lies in its intrinsic reward mechanism that maximizes mutual information between actions and policies, promoting a variety of behaviors even in homogeneous datasets.

**Strengths:**

see questions

**Weaknesses:**

see questions

**Questions:**

This paper proposes a new intrinsic reward system that maximizes the mutual information between actions and policies, encouraging behavioral diversity. Through the combination of EDAC method to estimate Q-value uncertainty, the proposed method achieves better performance than other baselines. However, the main concern is that the idea of introducing mutual information in offline RL is also proposed by one previous work [1]. Eq. 5 and 6 of [1] are similar to the objective function in this paper. The authors should claim the difference and novelty compared to [1] and consider involving it as a baseline in experiments.

[1] Ma, Xiao, et al. "Mutual Information Regularized Offline Reinforcement Learning." arXiv preprint arXiv:2210.07484 (2022). (accepted by neurips 2024)

---

> ### Author Response · Authors · 2024-11-22
> **Author Response**
>
> We thank reviewer BSoj for the insightful comments. Below, we address the questions raised by the reviewer. We hope the replies could help the reviewer further recognize our contributions. Thank you.
>
> **Q1. This paper proposes a new intrinsic reward system that maximizes the mutual information between actions and policies, encouraging behavioral diversity. Through the combination of EDAC method to estimate Q-value uncertainty, the proposed method achieves better performance than other baselines.  However, the main concern is that the idea of introducing mutual information in offline RL is also proposed by one previous work [1]. Eq. 5 and 6 of [1] are similar to the objective function in this paper. The authors should claim the difference and novelty compared to [1] and consider involving it as a baseline in experiments.**
>
> A1:
> In our model, we explicitly consider a multiple-policies model. This multi-policy perspective allows us to investigate the interactions and behavioral variances among different policies, which is critical for enhancing exploration beyond the dataset's confines. This stands in contrast to the approach taken by Ma, Xiao, et al., where the framework is built around a single-policy model. Their primary objective is to regularize the agent within the scope of existing data, focusing on reducing the uncertainty of actions given specific states without the need to account for interactions between different agent strategies.
> The two approaches also differ in their algorithmic formulations:
> - **Ma, Xiao, et al.'s Model:** They define the mutual information between the state $S$ and action $A$ as:
> $$I(S; A) = E_{p(s, a)}\left[ \log \frac{p(s, a)}{p(s)p(a)} \right] = E_{p(s, a)}\left[ \log \frac{p(a \mid s)}{p(a)} \right]$$
> focusing on minimizing the uncertainty of actions in given states, thereby aligning closely with historical data to reduce risks.
> - **Our Approach:** We define the mutual information between action $A$ and policy $\Pi$ in the context of state $S$ as:
> $$I(A ; \Pi | S) = E_{(a, \pi) \sim p(A, \Pi)} \left[ \log \frac{p(a | s, \pi)}{\sum_{i=1}^M p(a | s, \pi^i)p(\pi^i)} \right],$$
> where our focus is on maximizing the diversity among actions taken by different policies, highlighting the unique contributions each policy can offer to the decision-making process.
>
> Given these substantial differences in focus, methodology, and objectives, our work advances the domain of offline RL by exploring new avenues for action diversity and policy interaction. We acknowledge the foundational contributions of Ma, Xiao, et al. and will consider incorporating their model as a baseline in future experiments to further delineate the unique aspects of our approach.

---

> ### Author Response · Authors · 2024-11-25
> **Author Response**
>
> Dear reviewer BSoj,
>
> We appreciate the time and effort you have invested in reviewing our manuscript. If possible, could you kindly let us know if our responses have adequately addressed your questions? Please let us know if you have any further comments. We would be delighted to provide additional responses. Thank you.

---

### Official Review · Reviewer_8Awq · 2024-11-03

**Soundness:** 2
**Presentation:** 3
**Contribution:** 2
**Rating:** 6
**Confidence:** 4

**Summary:**

The paper tries to address a limitation in offline RL, where existing methods often focus on optimizing performance at the expense of promoting diverse behaviors. Traditional approaches that rely on well-constructed, heterogeneous datasets struggle with less diverse data. To overcome this, the authors propose a new intrinsic reward mechanism that encourages behavioral diversity regardless of the dataset's heterogeneity. This is achieved by maximizing the mutual information between actions and policies for each state, allowing agents to learn a variety of behaviors, even those not explicitly present in the data. To manage the risks associated with out-of-distribution actions, the authors incorporate the ensemble-diversified actor-critic (EDAC) method to estimate Q-value uncertainty, preventing suboptimal behavior adoption. Experiments using the D4RL benchmarks on MuJoCo tasks show that the proposed method successfully achieves behavioral diversity while maintaining performance across environments with both heterogeneous and homogeneous datasets.

**Strengths:**

1. The paper introduces a novel intrinsic reward mechanism that encourages behavioral diversity, a significant advancement over existing methods that often overlook this aspect.
2. Although there is some confusion regarding the details of the method, the writing logic used to introduce the approach and model is very clear and easy to follow.

**Weaknesses:**

There is some confusion about the method for me to understand the paper. Please refer to Questions.

**Questions:**

Overall, I have some questions:

Q1.  How are the heterogeneity and homogeneity of the dataset defined in the context of this study?

Q2. If I understand correctly, In Line205-208, $p^t(s_t|pi)$ is $p(s_t|s_{t-1},a_{t-1})$ and is nothing to do with $\pi$ (since you have specified \tau in $p(\tau|\pi)$, it can decompose to transition probability $p(s_t| s_{t-1}, a_{t-1})$ and $p(a_t|s_t, \pi)$ for t=0,1,...). Based on this, I think there may be some mistakes in the following analysis.

Q3. In practice, how to estimate $p(a_t|s_t)$ in Equation 8?


Q4. Can the experimental results be generalized to tasks with discrete action spaces, such as Atari, representing more complex environments? It would be advantageous if the methods could be validated more broadly.

Q5. The authors employ ensemble training in the experiments. Could we consider incorporating information between the policies to achieve final diversity, as done in [1]?

[1] Wu S et al. Quality-similar diversity via population based reinforcement learning[C]//The Eleventh International Conference on Learning Representations. 2023.

---

> ### Author Response · Authors · 2024-11-23
> **Author Response**
>
> We thank reviewer 8Awq for the insightful comments. Below, we address the questions raised by the reviewer. We hope the replies could help the reviewer further recognize our contributions. Thank you.
>
> **Q1. How are the heterogeneity and homogeneity of the dataset defined in the context of this study?**
>
> A1:
> In the context of our study, heterogeneity in a dataset is defined by the variety of user behaviors it contains. For example, in driving scenarios, heterogeneity is observed through diverse driving styles, such as some drivers preferring to closely follow the car ahead, while others frequently change lanes. This variance exemplifies heterogeneity. Conversely, homogeneity refers to the uniformity in user behaviors within the dataset, where most data points exhibit similar actions or decision-making patterns, often seen in scenarios where drivers show consistent driving styles.
> In our experiment, the Standard D4RL Dataset is derived from a Soft Actor-Critic (SAC) policy which generates homogeneous behavior patterns. In contrast, the Diverse D4RL Dataset is collected from a latent-conditioned policy, with various latent vectors, designed to capture a broader range of behaviors, hence more heterogeneous. As illustrated in Table 1, the baseline methods (DIVEOFF, CLUE, SORL) trained on heterogeneous datasets received high diversity scores; but their diversity dropped significantly when they were trained on homogeneous datasets. In contrast, our method achieved high diversity in both homogeneous and heterogeneous datasets.
>
> **Q2. If I understand correctly, In Line 205-208, $p^t(s_t|\pi^i)$ is  $p(s_t|s_{t−1},a_{t−1})$ and is nothing to do with $\Pi$ (since you have specified $\tau$ in $p(\tau|\pi)$, it can decompose to transition probability $p(s_t|s_{t−1},a_{t−1})$ and $p(a_t|s_t,\pi)$for t=0,1,...). Based on this, I think there may be some mistakes in the following analysis.**
>
> A2:
> We would like to clarify our notation regarding state transition probabilities. In our framework, $p^t(s_t | \pi^i) $ denotes the probability of agent $ \pi^i $ being in state $ s_t $ at time $ t $, conditioned on the entire sequence of states and actions from the start up to time $ t-1 $: $ s_0, a_0, \ldots, s_{t-1}, a_{t-1} $. This formulation allows for a more general expression than the typical one-step transition probability: $p^t(s_t | \pi^i) = p(s_t | s_{t-1}, a_{t-1}, \ldots, s_{t-k}, a_{t-k})$ where $k$ ranges from 2 to $ t $, depending on the depth of historical dependency considered.
> Notably, in the standard one-step Markov Decision Process (MDP) setting, this reduces to: $p(s_t | s_{t-1}, a_{t-1})$ and thus upholds Equation 8. Our formulation also holds in $k$-steps MDP settings, accommodating dependencies over multiple previous steps and also other historical conditioned transition probability. Therefore, our formula is applicable in scenarios involving more complex historical dependencies beyond the MDP framework.
>
> **Q3. In practice, how to estimate p(at|st) in Equation 8?**
>
> A3:
> Since $ p(a_t | s_t, \pi^i) = \pi^i(a_t | s_t) $, we have $ p(a_t | s_t) = \frac{1}{M} \sum_{i=1}^M \pi^i(a_t | s_t) $, where $ \pi^i(\cdot | s_t) $ is the agent $\pi^i$ and under the presumption of $ p(\pi^i) = \frac{1}{M} $ . Hence, $p(a_t|s_t)$ can be obtained exactly through forward passes of the policy networks.

---

> ### Author Response · Authors · 2024-11-23
> **Author Response**
>
> **Q4. Can the experimental results be generalized to tasks with discrete action spaces, such as Atari, representing more complex environments? It would be advantageous if the methods could be validated more broadly.**
>
> A4:
> We extended our evaluation to the Atari domain, where our comparison is primarily with SORL, as it is the only baseline method in our study that has also been tested on Atari environments. We executed the official SORL code available on their GitHub repository to ensure a fair comparison. For this part of our study, we set $\lambda$( the weight for  $U_{\text{Behavior}}$) to 1 We trained a model with three different policies and tested each across 10 trajectories using three random seeds to assess variability and consistency in performance.
> The results from these experiments indicate that our method not only performs well across various conditions in MuJoCo tasks but also extends effectively to other environments, including those with discrete action spaces, such as Atari.
>
> | Atari          | Performance         | State Diversity     | Action Diversity   |
> |----------------|---------------------|---------------------|--------------------|
> | **SpaceInvaders** |                     |                     |                    |
> | Ours           | 427.6 ± 50.2        | 0.64 ± 0.29         | 0.56 ± 0.15        |
> | SORL           | 422.6 ± 84.5        | 0.46 ± 0.25         | 0.27 ± 0.04        |
> | **Riverraid**     |                     |                     |                    |
> | Ours           | 1892.8 ± 309.7      | 0.57 ± 0.07         | 0.71 ± 0.16        |
> | SORL           | 1751.3 ± 313.1      | 0.32 ± 0.15         | 0.50 ± 0.13        |
>
> **Q5. The authors employ ensemble training in the experiments. Could we consider incorporating information between the policies to achieve final diversity, as done in [1]? [1] Wu S et al. Quality-similar diversity via population based reinforcement learning[C]//The Eleventh International Conference on Learning Representations. 2023.**
>
> A5:
> In Wu et al. In [1], diversity is engineered through the use of user-specified Behavior Descriptors, which promote varying agent behaviors to align with different user-defined criteria. In contrast, we use unique behavior (UB) to measure how unique the action is among agents, which allows for a more intrinsic exploration of diversity. However, integrating the concept from Wu et al. [1] into our framework could be beneficial for scenarios where specific types of diversity are desirable. By adopting their strategy of using Behavior Descriptors, we can tailor the diversity generated by our model to fit particular user needs or to ensure alignment with targeted goals.
>
> We conducted experiments (please refer to: https://imgur.com/a/e0TKFsS) on the Maze2d environment as depicted in Figure 2 of our main paper. We define $B(\pi_θ)$ in  Wu et al. [1]  as panel (a), referred to as the user-specified Behavior Descriptors. Panel (b), (c), and (d) demonstrate the outcomes of this setup. Panel (b) showcases the trajectory of the target agent, which closely follows the target behavior, highlighting the efficacy of our method in embedding and controlling specific agent behaviors. In contrast, Panel (c) and (d) depict the trajectories of other agents who were not given the matching bonus but were still subject to the dataset and unique behavior rewards. These agents exhibit diverse behaviors, diverging significantly from the target, thus emphasizing the diversity achievable under our framework.

---

> > ### Comment · Reviewer_8Awq · 2024-11-24
> > **Response to Authors**
> >
> > Thank you for addressing most of my questions. I have raised my score accordingly. However, regarding Q2, I am still confused about introducing $\pi^i$. For both one-step MDP and k-step MDP, the transition probability is independent of $\pi^t$.

---

> > > ### Author Response · Authors · 2024-12-01
> > > **Author Response**
> > >
> > > We appreciate the insightful feedback provided by Reviewer 8Awq. Below, we address the questions regarding $p^t(s_t|\pi^i)$ raised by the reviewer. **Please let us know if you have any further comments. We would be delighted to provide additional responses. Thank you.**
> > >
> > > Under the standard MDP framework, the transition probability $P(s_{t} | s_{t-1}, a_{t-1})$ is indeed independent of any specific policy $\pi$, applicable across both one-step and multi-step MDP scenarios. Building on this foundation, we further explain how $p^t(s_t|\pi^i)$ is designed to accommodate environments characterized by more intricate historical dependencies:
> > >
> > >
> > > ## 1. Motivation for Introducing $p^t(s_t|\pi^i)$
> > >
> > > Our decision to introduce the $p^t(s_t|\pi^i)$ representation in our paper is driven by a desire to characterize diversity for sequential decision making for **both the standard MDP framework and beyond**. We illustrate this with examples from two classical frameworks: Partially Observable MDP (POMDP) and Contextual MDP.
> > >
> > > ## 2. Example 1: Partially Observable MDP (POMDP)
> > >
> > > In the POMDP setting, as described by Zamboni et al., 2024, the agent only observes partial information about the true state through noisy measurements, represented by the observation function $O(o_t | s_t)$. A belief $b(s)$ about the state space is maintained due to the inherent uncertainty in state observations. The operator $T$ is utilized to update this belief based on new information where $b’ = T^{o,a}(b)$. Additionally, $i_t$ represents the information set available at time $t$, which may include both belief states and observations.  The trajectory probability is defined as: $p(\tau|\pi) = p(s_0) \prod_{t=0}^{T-1} O(o_t | s_t) \pi(a_t | i_t) P(s_{t+1} | s_t, a_t) T^{o_t,a_t}(b_{t+1} | b_t)$. Under the definition of the belief trajectory probability, $p^t(s_t|\pi^i)$ is then represented as:
> > >  $$p^t(s_t|\pi^i) = O(o_{t-1} | s_{t-1}) P(s_{t} | s_{t-1}, a_{t-1})T^{o_{t-1},a_{t-1}}(b_{t}^{\pi^i} | b_{t-1}^{\pi^i}).$$
> > > **Due to each agent maintaining its belief states models [Singh, Gautam, et al., 2021],  the variability in these belief states $b_t^{\pi^i}$ leads to diverse $p^t(s_t|\pi^i)$ values among the agents.**
> > >
> > >
> > > ## 3. Example 2: Contextual MDP
> > >
> > > In the contextual MDP framework, [Tennenholtz et al., 2023] offers a framework for modeling decision-making processes influenced by history-dependent contexts. These models incorporate states (S), actions (A), and contexts (X) with respective cardinalities. Over time, agents accumulate a history that details past states, actions, and contexts leading up to the current situation. The transition probability of contextual MDP is influenced by contextual factors $x_t$ that reflect historical long-term environmental impacts, $p^t(s_t \mid \pi^i) $ is represented by:
> > > $$
> > > p^t(s_t \mid \pi^i) = p(s_{t} | s_{t-1}, a_{t-1}, x_{t-1}^{\pi^i}).
> > > $$
> > > **Agents with varying historical experiences have different contextual influences $x_{t-1}^{\pi^i}$, which diversifies $p^t(s_t|\pi^i)$ among them.**
> > >
> > > ## 4. Generalization Across Environments
> > > The definitions of $p^t(s_t|\pi^i)$ in both POMDPs and contextual MDPs demonstrate how this measure can vary significantly among agents, which is pivotal for understanding and predicting agent behavior in more complex and diverse settings for both one-step and $k$-step cases. **Our goal is to have a general form of definition and extend experiments of the unique behavior framework to these and other environmental settings, enhancing the applicability and robustness of our approach.**
> > >
> > > ## 5. Fundamental Relationship Between Behavior and Path Uniqueness
> > >
> > > **In Inequation (8), behavior uniqueness serves as a lower bound for path uniqueness.** **This relationship holds true across all decision-making frameworks discussed, including standard MDP, POMDPs and contextual MDPs.**
> > >
> > > [Zamboni et al., 2024] Riccardo Zamboni et al., "How to explore with belief: State entropy maximization in POMDPs," International Conference on Machine Learning, 2024.
> > > [Singh, Gautam, et al., 2021] Gautam Singh et al., "Structured world belief for reinforcement learning in POMDP," International Conference on Machine Learning, 2021.
> > > [Tennenholtz et al. 2023] Guy Tennenholtz, et al., "Reinforcement learning with history dependent dynamic contexts." International Conference on Machine Learning, 2023.

---

> > > > ### Comment · Reviewer_8Awq · 2024-12-02
> > > > **Response**
> > > >
> > > > Thanks for the detailed illustration for introducing $p^t(s_t|\pi^i)$. This seems to be clearer and solves my confusion. Adding this to the appendix as the theoretical motivation for the method would be nice.

---

### Official Review · Reviewer_vcoR · 2024-11-03

**Soundness:** 3
**Presentation:** 3
**Contribution:** 2
**Rating:** 6
**Confidence:** 3

**Summary:**

This paper introduces a novel approach to offline reinforcement learning that promotes behavioral diversity through an intrinsic reward mechanism based on maximizing mutual information between actions and policies. The proposed method works effectively with homogeneous datasets, enabling agents to learn varied behaviors beyond those present in the training data. Moreover, the approach also incorporates uncertainty estimation through ensemble-diversified actor-critic (EDAC) to avoid OOD issues, demonstrating strong performance across both heterogeneous and homogeneous D4RL benchmark environments.

**Strengths:**

1. The proposed method effectively promotes policy diversity across both homogeneous and heterogeneous datasets, addressing a significant limitation in existing approaches.
2. The mutual infomation-based intrinsic reward is technically sound and intuitive.
3. The paper is very well-written and easy to follow.

**Weaknesses:**

1. I am not sure the illustrative example given in Figure 2 is the best scenario to demonstrate behavioral diversity as the demonstrated policies appear relatively similar despite attempting to showcase distinct behaviors.
2. The empirical evaluation's scope is limited, focusing on only three MuJoCo benchmark tasks, which may not fully validate the method's generalizability.
3. The theoretical transition from path diversity to behavior diversity, while conceptually reasonable, requires more justification, particularly regarding equivalence conditions and underlying assumptions.

**Questions:**

1. How does dataset quality impact the method's effectiveness in promoting behavioral diversity?
2. Could the authors provide a comprehensive hyperparameter analysis for $\lambda$ and $N$ to better understand their influence on performance?
3. How might the learned behavioral diversity translate to improved performance in downstream tasks, for example, few-shot adaptation?

---

> ### Author Response · Authors · 2024-11-22
> **Author Response**
>
> We thank reviewer vcoR for the insightful comments. Below, we address the questions raised by the reviewer. We hope the replies could help the reviewer further recognize our contributions. Thank you.
>
> **Q1.I am not sure the illustrative example given in Figure 2 is the best scenario to demonstrate behavioral diversity as the demonstrated policies appear relatively similar despite attempting to showcase distinct behaviors.**
>
> A1:
> In Figure 2, diversity refers to the differences in behavior among policies, not variations within the same policy across different trials. Each policy is represented by a unique trajectory pattern, demonstrating how it navigates the environment. For instance, Policy 2 tends to navigate in larger loops, often closer to the upper walls of the environment. In contrast, Policy 3 exhibits behavior characterized by tighter loops and a propensity to stay closer to the lower walls. The diversity we aim to illustrate is the distinct navigational strategies adopted by different policies when placed in the same starting conditions.
>
> **Q2. The empirical evaluation's scope is limited, focusing on only three MuJoCo benchmark tasks, which may not fully validate the method's generalizability.**
>
> A2:
> In our offline reinforcement learning experiments, the type of training data is a crucial factor for assessing model robustness against different datasets. We utilized both the Standard D4RL dataset and the Diverse D4RL dataset across three MuJoCo tasks, creating a total of 18 distinct experimental conditions categorized into medium-expert, medium-replay, and medium settings.
> In this rebuttal, we extended our evaluation to the Atari domain, where our comparison is primarily with SORL, as it is the only baseline method in our study that has also been tested on Atari environments. We executed the official SORL code available on their GitHub repository to ensure a fair comparison. For this part of our study, we set $\lambda$( the weight for  $U_{\text{Behavior}}$) to 1. We trained a model with three different policies and tested each across 10 trajectories using three random seeds to assess variability and consistency in performance.
> The results from these experiments indicate that our method not only performs well across various conditions in MuJoCo tasks but also extends effectively to other environments, including those with discrete action spaces, such as Atari.
>
> | Atari          | Performance         | State Diversity     | Action Diversity   |
> |----------------|---------------------|---------------------|--------------------|
> | **SpaceInvaders** |                     |                     |                    |
> | Ours           | **427.6 ± 50.2**        | **0.64 ± 0.29**        | **0.56 ± 0.15**        |
> | SORL           | 422.6 ± 84.5        | 0.46 ± 0.25         | 0.27 ± 0.04        |
> | **Riverraid**     |                     |                     |                    |
> | Ours           | **1892.8 ± 309.7**      | **0.57 ± 0.07**         | **0.71 ± 0.16**        |
> | SORL           | 1751.3 ± 313.1      | 0.32 ± 0.15         | 0.50 ± 0.13        |
>
> **Q3. The theoretical transition from path diversity to behavior diversity, while conceptually reasonable, requires more justification, particularly regarding equivalence conditions and underlying assumptions.**
>
> A3:
> The theoretical transition from path diversity to behavior diversity does not rely on any assumption. Specifically, by adapting the mutual information concept from Equation 4 to the context between the state $ \mathcal{S} $ and the policy $ \Pi $, we obtain $ I(\mathcal{S}; \Pi) = H(\mathcal{S}) - H(\mathcal{S}|\Pi) = \mathbb{E}_{(\pi, s) \sim p(\Pi, \mathcal{S})} \left[ \log \frac{p(s|\pi)}{p(s)} \right] $. Given the non-negative nature of mutual information, this formulation supports the validity of Inequation (8).

---

> ### Author Response · Authors · 2024-11-22
> **Author Response**
>
> **Q4. How does dataset quality impact the method's effectiveness in promoting behavioral diversity?**
>
> A4: Thank you for raising this important question. We interpret dataset quality here as referring to the capability of the behavior policy (e.g., Medium-Expert, Medium, Medium-Replay). If our understanding is incorrect, please kindly let us know.
>
> To better illustrate, we summarize the results from Table 1 in our paper into a concise table. Since Medium-Replay represents data collected during the training of a policy, it inherently includes samples from both strong and weak stages of the policy (or equivalently, different policies). This makes the diversity within the dataset more complex and less controllable. Therefore, for this discussion, we focus on the Medium-Expert and Medium datasets.
>
> For our proposed method, the diversity within the dataset has a more significant impact on the resulting policy’s behavioral diversity, while the dataset's quality has minimal influence. In contrast, for baseline methods, the dataset quality plays a more significant role in determining the behavioral diversity of the policies. Fundamentally, higher data quality tends to reduce behavioral diversity in baseline methods, as can be observed in the results for DIVEOFF and CLUE. Interestingly, SORL demonstrates the opposite trend, which may be attributed to its policy performance when applied to Diverse D4RL datasets.
>
> | Datasets                   | Metrics               | | Ours   | DIVEOFF | CLUE  | SORL | | Ours   | DIVEOFF | CLUE  | SORL  |
> |----------------------------|--------------------|----|--------|---------|-------|-----|--|--------|---------|-------|-------|
> |                            |                        | **Standard D4RL Dataset**| |       |       |       | **Diverse D4RL Dataset** | |       |       |       |
> | **Medium-Expert** |  Performance                    |  | 101.36 | 89.58   | 74.44 | 47.53| | 97.65  | 96.53   | 87.7  | 68.6  |
> | **Medium-Expert**    |    Diversity                 |   | 0.7    | 0.37    | 0.45  | 0.54 | | 0.91   | 0.33    | 0.7   | 0.81  |
> | **Medium**         | Performance                     |  | 85.66  | 54.91   | 56.41 | 40.11| | 90.96  | 90.34   | 77.02 | 69.66 |
> | **Medium**           | Diversity                      | | 0.71   | 0.36    | 0.56  | 0.52 | | 0.89   | 0.41    | 0.85  | 0.69  |
>
> **Q5. Could the authors provide a comprehensive hyperparameter analysis for $\lambda$ and $N$ to better understand their influence on performance?**
>
> A5:
> In our model, $\lambda$ serves as the weight for  $U_{\text{Behavior}}$, balancing the emphasis between reward optimization and diversity. A higher value of $\lambda$ might overly prioritize diversity at the expense of achieving optimal rewards. Throughout our experiments, we set $\lambda$ to 1.0 across all offline datasets. An exception was made for the medium-expert Walker2d environment in the Diverse D4RL dataset, where it was adjusted to 0.5.
> Regarding the parameter $N$, which represents the number of ensembles, its primary role is to mitigate the selection of highly uncertain actions by the policy. A smaller $N$ could potentially allow for the selection of such uncertain actions, negatively impacting performance. In our experiments, we adopted the same settings used in EDAC for $N$. Specifically, $N$ is set to 10 for the HalfCheetah and Walker tasks, and 50 for the Hopper task.
>
> **Q6. How might the learned behavioral diversity translate to improved performance in downstream tasks, for example, few-shot adaptation?**
>
> A6:
> The learned behavioral diversity in our model holds potential advantages for improving performance in downstream tasks, such as few-shot adaptation. This capability arises from the model's ability to experience behaviors outside the dataset, providing it with a better generalization ability to new tasks.

---

> > ### Comment · Reviewer_vcoR · 2024-11-24
> >
> > Thank you for the detailed response, particularly the additional experiments on the Atari benchmark and the explanation of data quality. The analysis of the differing impacts of dataset diversity and quality across algorithms is interesting. I will maintain my previous score and would be glad to see this work accepted.

---

### Meta-Review · Area_Chair_LPU9 · 2024-12-20

**Metareview:**

This paper presents a mechanism for enhancing behavioural diversity through the use of an intrinsic reward that aims to maximize mutual information between actions and policies at each state. Additionally, they incorporate a mechanism to estimate Q-value uncertainty to avoid suboptimal behaviours from out-of-distribution actions. The method is thoroughly evaluated empirically to show its performance.

Overall this paper is well-written, the idea is sound, and the experiments are effective at showing the strength of this method. However, there is a general consensus on the novelty and impact of the technical contribution itself. Given that the use of mutual information for increasing diversity is not particularly novel, stronger analyses that motivate its use in offline RL would be beneficial; in particular, there is a potential tradeoff between diversity and performance that is not well-explored and which would help situate this work better, and also help future researchers build on it. Adding a theoretical analysis of the proposed methods could help in this regard.

In conclusion, although this work has promise, I believe it is not quite ready for publication at ICLR. I encourage the authors to incorporate the feedback from the reviewers so as to strengthen their work for a future submission.

**Additional Comments On Reviewer Discussion:**

There were a number of clarifying questions from the reviewers which the authors properly addressed. Additionally, based on reviewer feedback, the authors ran extra experiments on Atari experiments.

---

### Decision · Program_Chairs · 2025-01-22

Reject